# Right for the Right Reasons: Avoiding Reasoning Shortcuts via Prototypical Neurosymbolic AI

**Luca Andolfi**
Imperial College London, Sapienza University of Rome
l.andolfi24@imperial.ac.uk

**Eleonora Giunchiglia**
Imperial College London
e.giunchiglia@imperial.ac.uk

## Abstract

Neurosymbolic AI is growing in popularity thanks to its ability to combine neural perception and symbolic reasoning in end-to-end trainable models. However, recent findings reveal these are prone to shortcut reasoning, i.e., to learning unindented concepts–or neural predicates–which exploit spurious correlations to satisfy the symbolic constraints. In this paper, we address reasoning shortcuts at their root cause and we introduce Prototypical Neurosymbolic architectures. These models are able to satisfy the symbolic constraints (*be right*) because they have learnt the correct basic concepts (*for the right reasons*) and not because of spurious correlations, even in extremely low data regimes. Leveraging the theory of prototypical learning, we demonstrate that we can effectively avoid reasoning shortcuts by training the models to satisfy the background knowledge while taking into account the similarity of the input with respect to the handful of labelled datapoints. We extensively validate our approach on the recently proposed `rsbench` benchmark suite in a variety of settings and tasks with very scarce supervision: we show significant improvements in learning the right concepts both in synthetic tasks (`MNIST-EvenOdd` and `Kand-Logic`) and real-world, high-stake ones (`BDD-OIA`). Our findings pave the way to prototype grounding as an effective, annotation-efficient strategy for safe and reliable neurosymbolic learning. [1]

## 1 Introduction

Neurosymbolic AI [36, 12] (NeSy) holds the promise of being able to combine the high adaptability of neural networks with the reasoning abilities of more traditional AI. This results in methods that are more interpretable [11, 4, 8], able to learn from fewer labelled datapoints [2, 49, 35] and are compliant by-design with the given constraints [1, 18]. However, recent studies [28, 29] have shown that even state-of-the-art NeSy methods can fall prey of *reasoning shortcuts*, i.e., (intuitively) spurious associations among the learnt concepts—a.k.a. the neural predicates [26]–that satisfy the given symbolic constraints and yet violate the intended semantics.

Some attempts have already been made to address the problem, however existing mitigation strategies carry practical trade-offs. For example, the solution proposed in [29] involves dense annotation of a high number of datapoints which can be costly. On the other hand, unsupervised solutions like training jointly with a reconstruction loss [29] or the Shannon entropy [27] work only in limited settings, but fail when applied in more challenging scenarios like those in `rs-bench` [5].

In this paper, we address reasoning shortcuts at their root cause and we introduce Prototypical Neurosymbolic architectures. Thanks to the prototypes, our mitigation strategy requires minimal annotations and delivers extremely positive results even in challenging scenarios like those in `rs-bench`. The work is based on the simple intuition that whenever we are updating the weights of our neural

---

[1]The code is available on our `r4rr` Github page

network, we need to take into account two possibly orthogonal factors: (i) the satisfaction of the background knowledge and adherence to the available ground truth labels, which NeSy methods naturally do, and (ii) the similarity of the input with respect to the handful of labelled datapoints. We show that if prototypical networks are used in conjunction with NeSy methods, then both factors are taken into account at each weights update step. On the contrary, when using NeSy methods in conjunction with standard neural networks, this cannot happen. We also show that under the assumption of clusterability in the embedding space, the number of deterministic shortcuts gets significantly reduced.

We extensively validate our approach on the recently proposed `rsbench` [5] benchmark suite in a variety of settings and tasks with very scarce supervision: we show significant improvements in learning the right concepts both in synthetic tasks (`MNIST-EvenOdd` and `Kand-Logic`) and real-world, high-stake ones (`BDD-OIA`). As an example, with just one labelled datapoint for each concept, we are able to increase the F1-score macro over the concepts for the standard DeepProbLog [26] from $4\%$ to $96\%$ in the `MNIST-EvenOdd` task and from $25\%$ to $94\%$ on the `Kand-Logic` task.

## 2    Problem Statement

**Notation.**    As customary, we denote scalar constants in lower-case $x$, random variables $X$ in upper case, and ordered sets of constants $\mathbf{x}$ and random variables $\mathbf{X}$ in bold typeface. We also will denote the set $[1, \dots n]$ as $[n]$ and the *prior knowledge* as K.

**Setting.**    In line with previous work [28, 29] we consider an underlying ground-truth data generation process $p^*(\mathbf{X}, \mathbf{Y}; \mathrm{K})$ having form as illustrated in Figure 1, where (i) $\mathbf{X}$ is the ordered set of random variables with domain $\mathcal{X}$ representing the datapoints in our learning problem, (ii) $\mathbf{Y}$ is the ordered set of $r$ discrete random variables with domain $\mathcal{Y}$ representing the ground truth labels in our learning problem, (iii) $\mathbf{G}$ is the ordered set of $k$ discrete random variables ranging in $\mathcal{G} = [h_1] \times \ldots \times [h_k]$, which influence the observed datapoint and determine the ground truth label, and (iv) $\mathbf{S}$ is a finite ordered set of random variables which influence the observed data but not the ground truth labels.

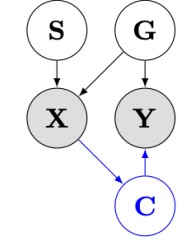

Figure 1: Ground truth generation process (in **black**).

A *NeSy predictor* is a model that infers the labels $\mathbf{Y}$ by reasoning over the background knowledge K and a set of $k$ discrete concepts $\mathbf{C}$ (taking values in $\mathcal{C}$ with the assumption that $\mathcal{C} = \mathcal{G}$) extracted from the sub-symbolic input $\mathbf{X}$. The background knowledge is assumed to be given beforehand.

**Example 1** *(`MNIST-Addition` [26]) In this classic problem, the pair of MNIST images $\mathbf{x} = ($▨$, $▨$)$ and their ground truth label $5$ is determined by the ground truth concepts $\mathbf{g} = (3, 2)$. The writing style $\mathbf{s}$ only influences the appearance of those digits. The ground truth concepts and the labels are compliant with the background knowledge $\mathrm{K} = (Y = G_1 + G_2)$.*

**Goal.** Our goal is to learn the parameters $\theta$ of a NeSy predictor $p_\theta(\mathbf{Y} \mid \mathbf{X}; \mathrm{K})$ such that $p_\theta(\mathbf{C} \mid \mathbf{x}) = p^*(\mathbf{G} \mid \mathbf{x})$, for all $\mathbf{x} \in \mathcal{X}$.

**The reasoning shortcut problem.** The reasoning shortcut problem arises because we do not have access to the ground truth $\mathbf{G}$ but only to a proxy of it, i.e., $\mathbf{Y}$.

Let $\mathcal{D} = \{(\mathbf{x}_i, \mathbf{y}_i)\}_{i=1}^n$ be a finite dataset, the we say that a NeSy predictor $p_\theta$ takes a *reasoning shortcut* (RS) if $p_\theta$ achieves maximal log-likelihood $\mathcal{L}$ on the training data but it does not match the ground truth distribution, i.e.,

$$\mathcal{L}(p_\theta, \mathcal{D}, \mathrm{K}) = \max_{\theta' \in \Theta} \mathcal{L}(p_{\theta'}, \mathcal{D}, \mathrm{K}) \quad \text{and} \quad p_\theta(\mathbf{C} \mid \mathbf{X}) \neq p^*(\mathbf{G} \mid \mathbf{X}), \tag{1}$$

where $\mathcal{L}(p_\theta, \mathcal{D}, \mathrm{K}) = \frac{1}{|\mathcal{D}|} \sum_{(\mathbf{x}, \mathbf{y}) \in \mathcal{D}} \log p_\theta(\mathbf{y} \mid \mathbf{x}; K)$. Observe every RS corresponds to a *deterministic optimum* for $\mathcal{L}$.

**Example 2** *(Ex. 1, Cont'd.) Consider a restriction of the dataset where the ground truth for all datapoints is either $\mathbf{g} = (0, 6)$ or $\mathbf{g}' = (2, 8)$. Assume $p_\theta(\mathbf{C} \mid \mathbf{G})$ is a deterministic distribution so that $p_\theta((5, 5) \mid (2, 8)) \approx 1.0$ and $p_\theta((3, 3) \mid (0, 6)) \approx 1.0$. A NeSy predictor $p_\theta$ whose (i) distribution $p_\theta(\mathbf{C} \mid \mathbf{G})$ is as shown before and (ii) likelihood $\mathcal{L}(p_\theta, \mathcal{D}, \mathrm{K})$ is maximal, takes a RS.*

# 3 Prototypical Neurosymbolic AI

Our goal is to learn the parameters $\theta$ of a NeSy predictor $p_\theta(\mathbf{Y} \mid \mathbf{X}; K)$ such that $p_\theta(\mathbf{C} \mid \mathbf{x}) = p^*(\mathbf{G} \mid \mathbf{x})$ for all $\mathbf{x} \in \mathcal{X}$. However, in absence of any labelled datapoints, the network might take reasoning shortcuts as there is nothing more than the loss over the labels $\mathbf{Y}$ to guide the network.

For ease of presentation, we first assume we have at least one labelled datapoint for each concept, and then we drop such assumption.

Given a finite dataset $\mathcal{D}$ and a concept $c$, let

$$\mathcal{S}_c = \{\mathbf{x} \mid (\mathbf{x}, \mathbf{y}) \in \mathcal{D}, c \in \mathbf{y}\} \tag{2}$$

be *support set for concept* $c$. This is the available set of datapoints which we can use to "anchor" the representation of each concept, hence avoiding the RS problem. For every concept $c$ we create a *centroid* or *prototype* $\mathbf{c}_c \in \mathbb{R}^m$ starting from $\mathcal{S}_c$ i.e., a representative vector where all datapoints belonging to concept $c$ should be mapped to. To this end, we use $k$ prototype extractors (one for each set of mutually exclusive concepts in $\mathcal{C}$). Each *prototype extractor* is an embedding function $f_\theta^i : \mathbb{R}^d \to \mathbb{R}^{m_i}$ for $i \in [k]$ with learnable parameters $\theta$ which allows us to compute the centroids:

$$\mathbf{c}_c = \frac{1}{|\hat{\mathcal{S}}_c|} \sum_{\mathbf{x} \in \hat{\mathcal{S}}_c} f_\theta^i(\mathbf{x}), \qquad \text{for } i \in [k], \tag{3}$$

where $\hat{\mathcal{S}}_c \subseteq \mathcal{S}_c$ is a randomly sampled subset of $\mathcal{S}_c$.

For every prototype extractor $f_\theta^i(\mathbf{x})$, we define a distance $\rho : \mathbb{R}^{m_i} \times \mathbb{R}^{m_i} \to [0, +\infty)$, and, for every new input $\mathbf{x}$, we decide to which concepts it belongs to by measuring its distance to each of the prototypes. In [42], it was shown that for a particular class of distance functions, called the *regular Bregman divergences family* [3], the prototypical networks algorithm is equivalent to performing mixture density estimation on the support set with an exponential family density. In this work we choose as distance the squared Euclidean distance—itself belonging to the Bregman divergences family—as it will allow us to meaningfully initialize the centroids of the unspecified classes. It is now possible to produce a distribution over the concepts based on a softmax over distances between $\mathbf{z}_i = f_\theta^i(\mathbf{x})$ and the prototypes of each class:

$$p_\theta(c \mid \mathbf{x}, c \in [h_i]) = \frac{\exp(-\|\mathbf{z}_i - \mathbf{c}_c\|_2^2)}{\sum_{c' \in [h_i]} \exp(-\|\mathbf{z}_i - \mathbf{c}_{c'}\|_2^2)} \qquad \text{for } i = [k]. \tag{4}$$

We train the prototype extractors to perform two tasks: (i) correctly classify the datapoints in the *query set for class* $c$: $\mathcal{Q}_c = \mathcal{S}_c \setminus \hat{\mathcal{S}}_c$ if available (i.e., $\mathcal{Q}_c \neq \emptyset$) and (ii) make predictions that are coherent with the background knowledge K for all the datapoints in the training set $\mathcal{D}$ which are not labelled with a concept. An overview of the training procedure can be seen in Algorithm 1.

We now relax the previous assumption that each concept must have at least one labelled datapoint, hence we can have a concept $c$ such that $\mathcal{S}_c = \emptyset$. Instead, we require only that, within each set of mutually exclusive concepts (i.e., for every $[h_i]$ with $i \in [k]$), there exist at least two datapoints labelled with two distinct concepts. This condition ensures that we can express the centroids of concepts without labelled examples as functions of the centroids of concepts with at least one labelled instance.

Let $c \in [h_i]$ for some $i \in [k]$ be a concept for which there are no labelled datapoints in the training dataset $\mathcal{D}$. Let $\mathcal{H}_i \subseteq [h_i]$ be the set of concepts for which there exists a labelled datapoint in $\mathcal{D}$. Then the centroid for $c$ can be computed as

$$\mathbf{c}_c = \boldsymbol{\mu}_{\mathcal{H}_i} + \epsilon, \quad \text{with} \quad \epsilon \sim \mathcal{N}\left(0, \frac{\max_{c \in \mathcal{H}} \|\boldsymbol{\mu}_{\mathcal{H}_i} - \mathbf{c}_c\|_2^2}{\chi_{m,p}^2} \mathbf{I}_m\right), \tag{5}$$

where $\boldsymbol{\mu}_{\mathcal{H}_i} = \frac{1}{|\mathcal{H}_i|} \sum_{c' \in \mathcal{H}_i} \mathbf{c}_{c'}$ represents the mean of the known centroids in $[h_i]$, and $\chi_{m,p}^2$ is the $p$-th lower-tail quantile of a $\chi^2$ distribution with $m$ degrees of freedom. Unless explicitly stated, the parameter $p$ is set to 0.99 in our experiments. This ensures that, for each class $c$, the newly initialized centroid $\mathbf{c}_c$ has probability 0.99 (over the random draw of $\epsilon$) of lying within the hyperball whose radius is given by the distance from $\boldsymbol{\mu}_{\mathcal{H}_i}$ to the farthest known centroid. This encourages separation

**Algorithm 1** Training episode loss computation. $l_i$ is the number of classes sampled per set $\mathcal{H}_i$ and training episode. $s_c$ (resp. $q_c$) is the number of support (resp. query) examples per concept $c$.

---

**Require:** (i) Dataset $\mathcal{D}$, (ii) support set $\mathcal{S}_c$ for every concept $c$.
**Ensure:** Output the loss $\mathcal{L}$ for a randomly generated training episode.
1: $\mathcal{L} \leftarrow 0$
2: **for** $i \in \{1, \dots, k\}$ **do**
3:     $\mathcal{E} \leftarrow \text{RANDOMSAMPLE}(\mathcal{H}_i, l_i)$         ▷ Select classes per episode and prototype extractor
4:     **for** $c$ in $\mathcal{E}$ **do**
5:         $\hat{\mathcal{S}}_c \leftarrow \text{RANDOMSAMPLE}(\mathcal{S}_c, s_c)$         ▷ Select support examples
6:         $\mathcal{Q}_c \leftarrow \text{RANDOMSAMPLE}(\mathcal{S}_c \setminus \hat{\mathcal{S}}_c, q_i)$         ▷ Select query examples
7:         $\mathbf{c}_c \leftarrow \frac{1}{s_i} \sum_{\mathbf{x} \in \hat{\mathcal{S}}_c} f_\theta^i(\mathbf{x})$         ▷ Compute centroids for labelled classes
8:         **for** $\mathbf{x}$ in $\mathcal{Q}_c$ **do**

9:
$$\mathcal{L} \leftarrow \mathcal{L} + \frac{1}{l_i q_i}\left[||\mathbf{z}_i - \mathbf{c}_c||_2^2 + \log \sum_{c' \in [h_i]} \exp(-||\mathbf{z}_i - \mathbf{c}_{c'}||_2^2))\right]$$

10:     **for** $c \in [h_i] \setminus \mathcal{H}_i$ **do**
11:         $\mathbf{c}_c \leftarrow \boldsymbol{\mu}_{\mathcal{H}_i} + \epsilon$         ▷ Compute centroids for unlabelled classes
12: **for** $(\mathbf{x}, \mathbf{y}) \in \mathcal{D}$ **do**
13:     $\mathcal{L} \leftarrow \mathcal{L} + \mathcal{L}^{\text{NeSy}}(\mathbf{y}, \text{K})$         ▷ If necessary for NeSy model, compute NeSy loss

---

among mutually exclusive classes, while reducing the risk that a zero-shot prototype is placed far from the region spanned by the known embeddings, which could otherwise lead to prototype collapse, where multiple concepts are mapped to a single, constraint-satisfying centroid.

In the remainder, we will call a NeSy predictor that uses prototype extractors as outlined above a *prototypical NeSy predictor*.

## 4   Reasoning over Distances

Thanks to the prototypes, given an input $\mathbf{x}$, for every $i \in [k]$ the embedding $\mathbf{z}_i$ is updated taking into consideration not only the degree of constraints violation but its similarity to each centroid.

**Theorem 4.1** *Let* $\mathbf{y}$ *be the output of a prototypical NeSy predictor reasoning over the knowledge* K *and the set of concepts* $\mathcal{C} = [h_1] \times \dots \times [h_k]$. *If the predictor is trained using* $\mathcal{L}^{\text{NeSy}}(\mathbf{y}, \text{K})$, *then for every* $i \in [k]$ *the error calculated with respect to the embedding* $\mathbf{z}_i$ *is equal to:*

$$\nabla_{\mathbf{z}_i} \mathcal{L}^{\text{NeSy}}(\mathbf{y}, \text{K}) = 2 \sum_{c \in [h_i]} \frac{\partial \mathcal{L}^{\text{NeSy}}}{\partial y_c} y_c \left(\mathbf{c}_c - \sum_{c' \in [h_i]} y_{c'} \mathbf{c}_{c'}\right). \tag{6}$$

Proof in Appendix A. Let us call the term $\sum_{c' \in [h_i]} \mathbf{c}_{c'} y_{c'}$ the *centre of belief for datapoint* $\mathbf{x}$ *and set of classes* $[h_i]$ of the model, as it is the average over all the prototypes representing the classes in $[h_i]$ weighted by the probability assigned to each class by the model. The embedding $\mathbf{z}_i$ is thus updated in a way that ensures that—in the next learning iteration—the centre of belief for the datapoint $\mathbf{x}$ is closer to the weighted average over the distances of the prototype of each class to the centre of belief. The weights take into account both: (i) the degree of constraint violation, as captured by the magnitude of the gradient $|\partial \mathcal{L}^{\text{NeSy}}/\partial y_c|$ and (ii) how close the embedding is to a prototype, as captured by the $y_c$ term. The embedding update thus allows us at each step to try to satisfy the constraints while also taking into account the similarity between each class prototype and the embedding of the datapoint.

Notice that the same does not happen when training standard neural networks with a NeSy loss. In that case, even if we have some labelled datapoints, for the unlabelled datapoints the update will obviously only depend on the error done with respect to the NeSy loss, thus creating a fertile ground for reasoning shortcuts (especially as the number of unlabelled datapoints grows).

We now show what the embedding update looks like in the specific case of the semantic loss [49]. Given an input $\mathbf{x}$ and its corresponding output $\mathbf{y}$ we can write NeSy semantic loss as:

$$\mathcal{L}^{\text{NeSy}}(\mathbf{y}, \mathbf{K}) = -\log \sum_{\nu \models \mathbf{K}} \prod_{c \in \bigcup_i [h_i]} (y_c)^{\nu(c)} (1 - y_c)^{1 - \nu(c)}, \tag{7}$$

where each $\nu : \bigcup_i [h_i] \to \{0, 1\}$ represents (with a slight abuse of notation) a boolean assignment of the concepts in $\bigcup_i [h_i]$. We say that an assignment $\nu$ *satisfies* the knowledge $\mathbf{K}$ expressed as a propositional logic formula, denoted by $\nu \models \mathbf{K}$, if and only if the recursive evaluation of formula $\mathbf{K}$ under assignment $\nu$ yields 1. For every $c \in \bigcup_i [h_i]$ we define an independent Bernoulli random variable $Y_c \sim \text{Bernoulli}(y_c)$. This induces a distribution over full truth assignments $\nu : \bigcup_i [h_i] \to \{0, 1\}$. Then, we can rewrite the semantic loss as: $\mathcal{L}^{\text{NeSy}}(\mathbf{y}, \mathbf{K}) = -\log \sum_{\nu \models \mathbf{K}} p(\nu \mid \mathbf{y})$.

**Corollary 4.2** *Let $\mathbf{y}$ be the output of a prototypical NeSy predictor reasoning over the knowledge $\mathbf{K}$ and the set of concepts $\mathcal{C} = [h_1] \times \ldots \times [h_k]$. If the predictor is trained using semantic loss $\mathcal{L}^{\text{NeSy}}(\mathbf{y}, \mathbf{K})$ as defined in Equation 7, then for every $i \in [k]$ the error calculated with respect to the embedding $\mathbf{z}_i$ is equal to:*

$$\nabla_{\mathbf{z}_i} \mathcal{L}^{\text{NeSy}}(\mathbf{y}, \mathbf{K}) = 2 \sum_{c \in [h_i]} \left[ \frac{y_c - \mathbb{E}[Y_c \mid \mathbf{y}, \nu \models \mathbf{K}]}{y_c(1 - y_c)} \right] y_c \left( \mathbf{c}_c - \sum_{c'} \mathbf{c}_{c'} y_{c'} \right). \tag{8}$$

Proof in Appendix A. The above theorem shows that the embedding is moved closer to each centroid $\mathbf{c}_c$ by a value proportional to (i) the distance between $\mathbf{z}_i$ and $\mathbf{c}_c$ and (ii) to the difference between $y_c$ and the expected value of $Y_c$ conditioned on the satisfaction of the knowledge.

**Example 3** *Suppose $\mathcal{C} = [h_1] \times [h_2]$ where $[h_1]$ represents the concepts $\{Dog, Cat, Duck, Shark\}$ and $[h_2]$ represents the concepts $\{Mammal, Bird, Fish\}$. Let $\mathbf{K} = (Dog \lor Cat \to Mammal) \land (Duck \to Bird) \land (Shark \to Fish)$. Suppose we have an image $\mathbf{x}$ of a dog, whose embedding $\mathbf{z}_1$ is represented by ✖ in Figure 2. $\mathbf{z}_1$'s squared distances from the centroids are equal to the numbers next to the dotted lines. The centre of belief for $\mathbf{x}$ and $[h_1]$ is represented as ⭐. Let $\mathbf{z}_2$ have distance 2, 3, 4 from the prototypes of Mammal, Bird and Fish, respectively. The magnitude of the error computed with semantic loss by our method wrt Dog, Cat, Duck and Shark is $\sim 2.0$, $\sim 0.3$, $\sim 0.25$ and $\sim 5 \times 10^{-4}$, thus mirroring the importance that each centroid*

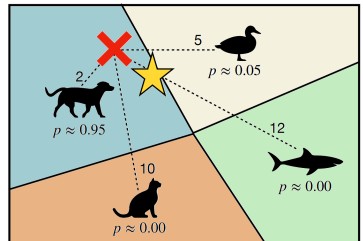

Figure 2: Prototype configuration for Example 3.

*has to the class prediction. On the contrary, if we were to use a standard neural network with semantic loss we would get that the errors for each of the above outputs all have magnitude of $\sim 1.0$.*

## 5 Counting Deterministic Reasoning Shortcuts

We now analyse the number of possible deterministic RSs admitted by a prototypical NeSy predictor. To isolate the structural sources of shortcuts, we work in an idealized setting where we assume that [**A1**] the distribution $p^*(\mathbf{X} \mid \mathbf{G}, \mathbf{S})$ is induced by a map $\phi \colon (\mathbf{g}, \mathbf{s}) \mapsto \mathbf{x}$, where $\phi$ is invertible and smooth over $\mathbf{s}$, [**A2**] the ground-truth process is consistent with the prior knowledge $\mathbf{K}$: the distribution $p^*(\mathbf{Y} \mid \mathbf{G}; \mathbf{K})$ is induced by a deterministic mapping $\beta_{\mathbf{K}} \colon \mathbf{g} \to \mathbf{y}$ and $p^*(\mathbf{y} \mid \mathbf{g}; \mathbf{K}) = 0$, for all $(\mathbf{g}, \mathbf{y})$ violating $\mathbf{K}$, and [**A3**] we have separability in the embedding space. This means that given a bijective assignment $\sigma \colon \mathcal{G} \to \mathcal{G}$ such that $\sigma(\mathbf{g})_i \in [h_i]$ with $i = [k]$, it holds that for every datapoint $\mathbf{x} = \phi(\mathbf{g}, \mathbf{s})$

$$\arg \min_{j \in [h_i]} || f_\theta^i(\mathbf{x}) - \mathbf{c}_j ||_2^2 = \sigma(\mathbf{g})_i. \tag{9}$$

Intuitively, all datapoints associated to the same concept $g \in [h_i]$, will have the embedding $f_\theta^i(\mathbf{x})$ closest to a centroid $\mathbf{c}$, but this centroid might be labelled with the wrong concept, i.e., it is not necessary that $g = \sigma(\mathbf{g})_i$.

Observe that, under [**A1**], each $p_\theta(\mathbf{C} \mid \mathbf{G})$ is induces exactly one mapping $\alpha : g \mapsto c$, such that $p_\theta(\mathbf{C} \mid \mathbf{G}) = \mathbb{1}\{\mathbf{C} = \alpha(\mathbf{G})\}$. We denote as $\mathcal{A}$ the set of all mappings $\alpha$. As shown in [29], the cardinality of $\mathcal{A}$ can be reduced leveraging disentangled concept extractor architectures: when a

model is *disentangled* then $p_\theta(\mathbf{C} \mid \mathbf{G})$ factorizes as $\prod_{i=1}^{k} p_\theta(C_i \mid G_i)$. In prototypical NeSy models the conditional distribution $p_\theta(\mathbf{Y} \mid \mathbf{X}; \mathrm{K})$ is induced by $k$ prototype extractors with embedding function $f_\theta^i \colon \mathbb{R}^d \to \mathbb{R}^{m_i}$, for every $i \in [k]$: under independence assumption, then our design enforces disentanglement *by construction*.

**Proposition 5.1** *Consider a prototypical NeSy predictor inducing $p_\theta(\mathbf{Y}|\mathbf{X}; \mathrm{K})$. Assume $\mathcal{G} = [h_1] \times \ldots \times [h_k]$ and let $\mathcal{H}_i \subseteq [h_i]$ be the set of concepts with at least one labelled datapoint from a dataset $\mathcal{D}$. Under [A1,2,3] the number of deterministic optima for $p_\theta(\mathbf{C} \mid \mathbf{G})$ is $p_\theta(\mathbf{C} \mid \mathbf{G})$ is*

$$\sum_{\alpha \in \mathcal{A}} \mathbb{1}\Big( \bigwedge_{\mathbf{g} \in \mathrm{supp}(\mathbf{G})} \Big( \bigwedge_{i \in [k]} \bigwedge_{g_j \in \mathcal{H}_i} (\alpha(\mathbf{g})_i = g_i \wedge g_i = g_j) \Big) \wedge \Big( (\beta_{\mathrm{K}} \circ \alpha)(\mathbf{g}) = \beta_{\mathrm{K}}(\mathbf{g}) \Big) \Big),$$

*$\alpha$ being the mapping induced by $p_\theta(\mathbf{C} \mid \mathbf{G})$ and $\mathcal{A}$ being the set of all possible mappings $\alpha$.*

Remarkably, by leveraging [**A3**], we achieve the same number of RSs obtained by the dense data-mitigation strategy proposed in [29]. This assumes that every supervised concept receives annotations for *every datapoint* in which it occurs. Hence, thanks to the prototypes, the labelling effort can be substantially reduced. Indeed, Proposition 5.1 implicitly assumes that for every $i \in [k]$ we have *one* labelled datapoint per class $g_i \in [h_i]$.

# 6 Experimental Analysis

**Metrics.** We evaluate the models coherently with [28, 29, 5] and we distinguish between the classes that are labelled as "*final labels*" and as "*concepts*". We measure the accuracy and F1-score wrt. both, denoted resp. as Acc(Y), F1(Y), Acc(C) and F1(C), and the concept collapse Cls(C).

**Tasks.** (i) `MNIST-EvenOdd` [28, 30]: is a variant of `MNIST-Addition` with restricted support. Each input is a pair of handwritten digits (concepts) labelled with their sum (final label). (ii) `Kand-Logic` [34, 30]: features the shape ($\square$, $\bigcirc$, $\triangle$) and color (red, yellow, blue) of geometric primitives (concepts), possibly defining a pattern to be predicted (final label). (iii) `BDD-OIA` [50]: an autonomous driving task proposing frames labelled with the actions to be undertaken by the ego-vehicle (final labels). The goal is to correctly classify the reasons (concepts) behind each action.

**Models.** For all tasks, we consider *DeepProbLog* [26] (DPL), *Logic Tensor Networks* [2] (LTN) and *Semantic Loss* [49] (SL), as implemented in [5], and we integrate them with prototypes. For `MNIST-EvenOdd`, we also consider *Coherent by Construction Networks* (CCN$^+$) [18].

**RQ1:** *Can Prototypical Neurosymbolic AI avoid Reasoning Shortcuts?*

To answer RQ1, we randomly label *just one* image per concept class in `MNIST-EvenOdd` and `Kand-Logic`. We then train each Prototypical NeSy model (SL+PNet, LTN+PNet, DPL+PNet) with support sets of size 10 and 9 for `MNIST-EvenOdd` and `Kand-Logic`, respectively. For `BDD-OIA` we annotate 126 images—almost the size of one training batch—with 21-dimensional binary vectors. We compare our results against: (i) the standard NeSy baselines (SL, LTN, and DPL), and (ii) the same models whose backbones are pretrained on annotated support sets enriched with ***over 150 additional datapoints*** obtained via data augmentation for

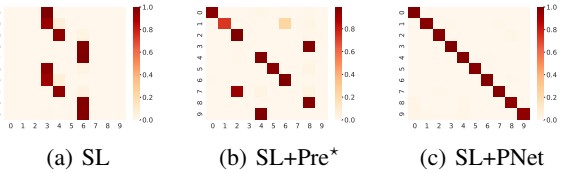

|   (a) SL   |  (b) SL+Pre$^\star$  |  (c) SL+PNet  |

Figure 3: `MNIST-EvenOdd` concept confusion matrices.

both `MNIST-EvenOdd` and `Kand-Logic`, as detailed in Appendix E (denoted as SL+Pre$^\star$, LTN+Pre$^\star$, and DPL+Pre$^\star$). No data augmentations were used for `BDD-OIA`. This second baseline demonstrates that, even on a simple dataset such as `MNIST-EvenOdd`, the use of prototypical networks can substantially outperform extensive data augmentation. Conversely, for `Kand-Logic`, the performance gap is minimal, which can be attributed to the high homogeneity of the concepts in this dataset. The results for `MNIST-EvenOdd` and `Kand-Logic` are shown in Tables 1 and 2. We observe that Prototypical NeSy models greatly outperform the standard baselines in the `MNIST-EvenOdd` task: for example LTN's Acc(C) jumps from $13\%$ to $95\%$ while Cls(C) drops from $80\%$ to $0\%$. The enhanced concept alignment is even more evident comparing the confusion matrices in Figure 3 for the SL model. While on `Kand-Logic` the gain wrt. concept metrics may appear less stark due to their homogeneity,

Table 2: Concept scores on `Kand-Logic`.

| | Acc(C)($\uparrow$) | F1(C)($\uparrow$) | Cls(C)($\downarrow$) |
|---|---|---|---|
| LTN | $0.36_{\pm0.06}$ | $0.22_{\pm0.06}$ | $0.64_{\pm0.16}$ |
| LTN+Pre$^\star$ | $0.90_{\pm0.01}$ | $0.90_{\pm0.01}$ | $0.00_{\pm0.00}$ |
| **LTN+PNet** | $\mathbf{0.91}_{\pm0.01}$ | $\mathbf{0.90}_{\pm0.01}$ | $\mathbf{0.00}_{\pm0.00}$ |
| SL | $0.35_{\pm0.05}$ | $0.20_{\pm0.04}$ | $0.73_{\pm0.12}$ |
| SL+Pre$^\star$ | $0.91_{\pm0.01}$ | $0.90_{\pm0.01}$ | $0.00_{\pm0.00}$ |
| **SL+PNet** | $\mathbf{0.92}_{\pm0.02}$ | $\mathbf{0.92}_{\pm0.02}$ | $\mathbf{0.00}_{\pm0.00}$ |
| DPL | $0.39_{\pm0.05}$ | $0.25_{\pm0.06}$ | $0.66_{\pm0.18}$ |
| DPL+Pre$^\star$ | $0.92_{\pm0.04}$ | $0.91_{\pm0.04}$ | $0.00_{\pm0.00}$ |
| **DPL+PNet** | $\mathbf{0.94}_{\pm0.00}$ | $\mathbf{0.94}_{\pm0.00}$ | $\mathbf{0.00}_{\pm0.00}$ |

Table 1: Concept scores on `MNIST-EvenOdd`.

| | Acc(C)($\uparrow$) | F1(C)($\uparrow$) | Cls(C)($\downarrow$) |
|---|---|---|---|
| LTN | $0.13_{\pm0.07}$ | $0.05_{\pm0.04}$ | $0.79_{\pm0.08}$ |
| LTN+Pre$^\star$ | $0.51_{\pm0.22}$ | $0.39_{\pm0.25}$ | $0.45_{\pm0.23}$ |
| **LTN+PNet** | $\mathbf{0.95}_{\pm0.03}$ | $\mathbf{0.95}_{\pm0.03}$ | $\mathbf{0.00}_{\pm0.00}$ |
| SL | $0.08_{\pm0.12}$ | $0.04_{\pm0.04}$ | $0.68_{\pm0.04}$ |
| SL+Pre$^\star$ | $0.52_{\pm0.25}$ | $0.42_{\pm0.28}$ | $0.40_{\pm0.24}$ |
| **SL+PNet** | $\mathbf{0.98}_{\pm0.01}$ | $\mathbf{0.98}_{\pm0.01}$ | $\mathbf{0.00}_{\pm0.00}$ |
| DPL | $0.08_{\pm0.10}$ | $0.04_{\pm0.05}$ | $0.67_{\pm0.05}$ |
| DPL+Pre$^\star$ | $0.64_{\pm0.22}$ | $0.55_{\pm0.28}$ | $0.32_{\pm0.24}$ |
| **DPL+PNet** | $\mathbf{0.97}_{\pm0.03}$ | $\mathbf{0.96}_{\pm0.04}$ | $\mathbf{0.00}_{\pm0.00}$ |
| CCN$^+$ | $0.03_{\pm0.07}$ | $0.01_{\pm0.03}$ | $0.73_{\pm0.05}$ |
| CCN$^+$+Pre$^\star$ | $0.33_{\pm0.11}$ | $0.22_{\pm0.10}$ | $0.57_{\pm0.09}$ |
| **CCN$^+$+PNet** | $\mathbf{0.96}_{\pm0.04}$ | $\mathbf{0.95}_{\pm0.05}$ | $\mathbf{0.00}_{\pm0.00}$ |

Table 3: Concept scores on `BDD-OIA`.

| | F1(C)($\uparrow$) | Cls(C)($\downarrow$) |
|---|---|---|
| DPL | $0.08_{\pm0.13}$ | $0.82_{\pm0.06}$ |
| DPL+Pre$^\star$ | $0.05_{\pm0.13}$ | $0.85_{\pm0.04}$ |
| **DPL+PNet** | $\mathbf{0.16}_{\pm0.15}$ | $\mathbf{0.35}_{\pm0.11}$ |

it carries significant improvements for the right prediction of the final label as shown in Table 10 in Appendix F, thus indicating a strong synergy between the symbolic and the neural components. Lastly, the superior ability of prototypical models to mitigate reasoning shortcuts (RSs) is also evident on `BDD-OIA`. As shown in Table 3, our PNet model reduces Cls(C) by more than half compared to both baselines, while achieving an F1(C) score that is more than twice as high as the best performing baseline. A more detailed analysis of RQ1 is provided in Appendix F.

**RQ2:** *How does Prototypical NeSy compare with other mitigation strategies?*

We consider the three mitigation strategies, and combinations thereof, as proposed in [27] and [29]. These are: (i) *Shannon-entropy regularization* (H): regularizing the models using the Shannon-entropy, (ii) *partial loss concept supervision* (C): taking a subset of the concepts and labelling all the corresponding datapoints in the dataset—following [29], we label the concepts "4" and "9", (iii) *reconstruction loss* (R): introducing an encoder-decoder architecture and training the model to jointly minimise $\mathcal{L}^{\text{NeSy}}$ and the reconstruction loss wrt the input. We carry out the comparison with our Prototypical NeSy models on the `MNIST-EvenOdd` task under the same settings for RQ1. The results obtained with each of these strategies and their combination are shown in Table 4. Crucially, each of the mitigation strategies in isolation does not manage to curb the RS problem, even with a costly solution like labelling all the datapoints where concepts "4" and "9" appear through C. Only by using them all together we get comparable results for DPL and LTN, while only F1(C)= 0.35 for SL.

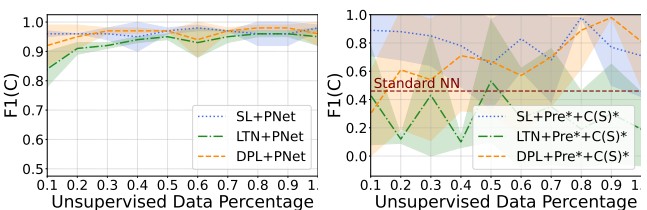

Figure 4: F1(C) on `MNIST-EvenOdd` across unlabelled data ratios.

**RQ3:** *Are Prototypical NeSy models robust wrt. the number of unlabelled datapoints?*

In Section 4 we show how Prototypical NeSy models update their parameters accounting at the same time both for the background knowledge and the proximity to each class centroid. With this RQ we validate the extent to which this update makes them less sensitive than traditional NNs to the amount of weakly supervised datapoints used at training time. To this end, we create 10 training subsets of `MNIST-EvenOdd` containing the original support sets together with a varying percentage of unlabelled examples (from 10% to 100% with a step increase of 10%). Then we train on these

Table 4: F1(C) and Cls(C) of different mitigations on `MNIST-EvenOdd`. Best results in bold.

| | F1(C)(↑) | | | Cls(C)(↓) | | |
| | DPL | SL | LTN | DPL | SL | LTN |
|---|---|---|---|---|---|---|
| + R | $0.07_{\pm 0.00}$ | $0.01_{\pm 0.00}$ | $0.05_{\pm 0.02}$ | $0.70_{\pm 0.00}$ | $0.70_{\pm 0.00}$ | $0.70_{\pm 0.00}$ |
| + H | $0.01_{\pm 0.00}$ | $0.01_{\pm 0.00}$ | $0.55_{\pm 0.14}$ | $0.70_{\pm 0.00}$ | $0.70_{\pm 0.00}$ | $0.08_{\pm 0.02}$ |
| + C | $0.01_{\pm 0.00}$ | $0.22_{\pm 0.01}$ | $0.12_{\pm 0.00}$ | $0.70_{\pm 0.00}$ | $0.60_{\pm 0.01}$ | $0.71_{\pm 0.00}$ |
| + R+H | $0.09_{\pm 0.02}$ | $0.01_{\pm 0.00}$ | $0.61_{\pm 0.10}$ | $0.63_{\pm 0.00}$ | $0.70_{\pm 0.00}$ | $0.21_{\pm 0.16}$ |
| + R+C | $0.01_{\pm 0.00}$ | $0.46_{\pm 0.07}$ | $0.10_{\pm 0.00}$ | $0.70_{\pm 0.00}$ | $0.47_{\pm 0.00}$ | $0.70_{\pm 0.00}$ |
| + H+C | $0.96_{\pm 0.06}$ | $0.31_{\pm 0.07}$ | $\mathbf{0.98}_{\pm 0.00}$ | $0.03_{\pm 0.05}$ | $0.50_{\pm 0.00}$ | $0.01_{\pm 0.00}$ |
| + R+H+C | $\mathbf{0.98}_{\pm 0.00}$ | $0.35_{\pm 0.10}$ | $0.95_{\pm 0.00}$ | $0.02_{\pm 0.00}$ | $0.50_{\pm 0.01}$ | $0.04_{\pm 0.00}$ |
| + PNet | $0.96_{\pm 0.04}$ | $\mathbf{0.98}_{\pm 0.01}$ | $0.95_{\pm 0.03}$ | $\mathbf{0.00}_{\pm 0.00}$ | $\mathbf{0.00}_{\pm 0.00}$ | $\mathbf{0.00}_{\pm 0.00}$ |

subsets (i) our prototypical models and (ii) baseline models whose backbone is pretrained using the support sets (Pre⋆, c.f. RQ1) and receiving additional supervision from them during training (C(S)⋆).

We report the comparison in Figure 4 wrt. F1(C). The Figures show that models employing traditional concept extractors struggle in effectively exploit the knowledge at hand. Moreover, they are particularly unstable and sensitive to both the network initialization and the number of unlabelled datapoints. On the flip side, prototypical ones use the background knowledge to refine their concept understanding and converge to stable solutions as more unlabelled data become available.

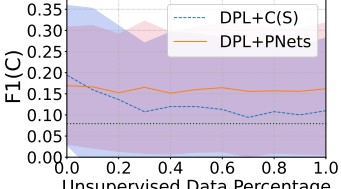 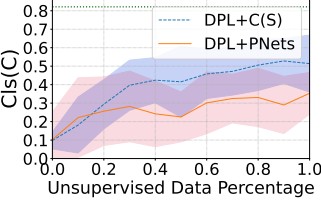

Figure 5: F1(C) and Cls(C) on `BDD-OIA` across different percentages of unlabelled datapoints. The green dotted line represents the baseline non-supervised DPL.

We run a similar experiment for `BDD-OIA` as well, where we consider a standard DPL model employing on extractor per concept that replicates the same disentanglement granularity of the prototypical counterpart. This model is also supervised through the support set. Figure 5 left (resp. right) shows that, as the number of unlabelled datapoints grows, the F1(C) (resp. Cls(C)) for standard DPL greatly decreases (resp. increases). On the contrary, the prototypical model is robust wrt. to this percentage and it consistently outperforms in both metrics the standard model.

**RQ4:** *What is the impact of having totally unlabelled classes?*

Ideally, one would like to have at least one labelled example for each class. In this RQ we test our approach for cases in which such assumption does not hold and we initialize the centroids for the unknown classes as in Equation 5. We also test our method with different numbers of totally unlabelled classes, running each experiment 5 times and randomly sampling each time the classes to leave unlabelled. In addition, for RQ4 and RQ5, we increase the support sets variety via standard augmentation techniques, as we found our models to perform slightly better in this case. The results obtained on `MNIST-EvenOdd` in terms of

Table 5: F1(C) when having different numbers of concepts left unspecified on `Kand-Logic`.

| # Unspec. | DPL | SL | LTN |
|---|---|---|---|
| 0 | $0.94_{\pm 0.00}$ | $0.92_{\pm 0.02}$ | $0.90_{\pm 0.01}$ |
| 1 | $0.65_{\pm 0.03}$ | $0.73_{\pm 0.03}$ | $0.64_{\pm 0.02}$ |
| 2 | $0.42_{\pm 0.02}$ | $0.55_{\pm 0.04}$ | $0.41_{\pm 0.02}$ |
| Baseline | $0.25_{\pm 0.06}$ | $0.22_{\pm 0.04}$ | $0.22_{\pm 0.06}$ |

F1(C) and F1(Y) are shown in Figure 6 and those on `Kand-Logic` in terms of F1(C) in Table 5. As expected, from Figure 6 and Table 5 we notice the more classes we are able to specify (i.e., provide examples for) the better the results. Crucially, even with minimal labels provided (e.g. two/four labelled images) Prototypical NeSy methods mitigate RSs.

**RQ5:** *How important is the centroid initialization for unlabelled classes?*

As explained in Section 4, it is important to make sure that the centroids for the unlabelled

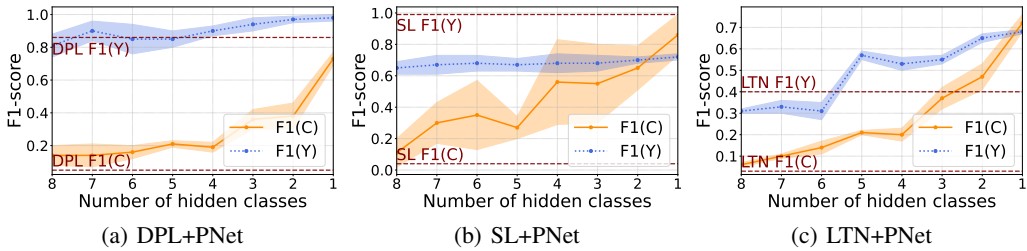

|  (a) DPL+PNet | (b) SL+PNet | (c) LTN+PNet |

Figure 6: F1(C) and F1(Y) obtained with Prototypical NeSy models trained on `MNIST-EvenOdd` when varying the number of unlabelled classes. The red lines indicate the performance of the corresponding baseline model.

classes lie among the labelled ones. In this ablation study, we show what happens if this assumption is violated. We run these experiments on the `MNIST-EvenOdd` dataset while picking different values of $p$, i.e., for the lower-tail quantiles of the $\chi^2$ distributions. The F1(C) results obtained with SL for $p = 0.2$, $p = 0.5$ and $p = 0.99$ are reported in Figure 7 and those in terms of Acc(C) and Cls(C) in Appendix F. We clearly see that as the value of $p$ increases the network performs better and better. Interestingly, we register the biggest difference in performance with 4 hidden classes, i.e., when the impact of the unlabelled centroids is high, and the known concepts provide enough information to allow for a meaningful computation of the centroids of the unlabelled classes.

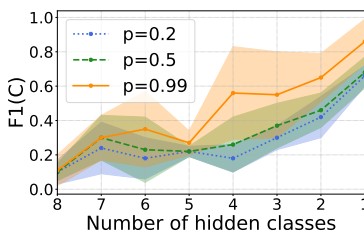

Figure 7: F1(C) with SL for $p = 0.2, 0.5$ and $0.9$.

## 7 Related Work

NeSy encompasses a diverse set of approaches that integrate symbolic reasoning with neural networks [36, 16, 12, 17]. A central challenge in this field lies in reconciling discrete, non-differentiable symbolic reasoning with the gradient-based optimization underlying neural networks. To enable end-to-end training, many NeSy approaches soften the semantics of logical reasoning via probabilistic [26, 51, 23, 32, 1, 46, 41, 24, 48, 9] or fuzzy logic frameworks [10, 11, 45, 2, 40]. A common strategy involves compiling background knowledge into the loss function to encourage constraint satisfaction during training [49, 13, 2], though this typically offers no guarantee that constraints will hold at inference time. In contrast, methods that restructure the model's output space can ensure constraint satisfaction by design [26, 14, 1, 7, 33, 18, 15, 19, 43, 44]. Despite the many advances [25, 47], [28, 29] showed that most NeSy systems are prone to RSs. Some mitigation strategies have been proposed [27, 29], but they can be either costly or they might fail in challenging settings like those proposed in `rs-bench`. To obviate to such problems, an interesting alternative direction has been taken by [30], where the authors do not try to avoid the RSs problem, but rather inform the user of when the model is taking a shortcut and facilitates acquiring informative dense annotations for mitigation purposes. For a comprehensive overview over the RSs problem in NeSy see [31].

## 8 Conclusion

In this paper, we introduce Prototypical NeSy models as a novel strategy for achieving more accurate and robust NeSy learning under very limited supervision. In particular, we showed how to effectively avoid reasoning shortcuts by anchoring neural embeddings to prototypes, which allows to exploit a handful of labelled examples to train prototype-based concept representations, that then get continuously refined via the symbolic knowledge. We show the precise dynamics of the embedding update behind our models' training and the difference with standard NeSy architectures. Moreover, we provide a theoretical analysis on the number of shortcuts possibly affecting our models. We validate them over a large set of experiments on the `rsbench` benchmark suite, where we show remarkable improvements, both in the synthetic tasks (`MNIST-EvenOdd` and `Kand-Logic`) and real-world high-stake ones (`BDD-OIA`).

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

# A  Proof Theorem 4.1

**Statement.**  Let $\mathbf{y}$ be the output of a prototypical NeSy predictor reasoning over the knowledge K and the set of concepts $\mathcal{C} = [h_1] \times \ldots \times [h_k]$. If the predictor is trained using $\mathcal{L}^{\text{NeSy}}(\mathbf{y}, \text{K})$, then for every $i \in [k]$ the error calculated with respect to the embedding $\mathbf{z}_i$ is equal to:

$$\nabla_{\mathbf{z}_i} \mathcal{L}^{\text{NeSy}}(\mathbf{y}, \text{K}) = 2 \sum_{c \in [h_i]} \frac{\partial \mathcal{L}^{\text{NeSy}}}{\partial y_c} y_c \left( \mathbf{c}_c - \sum_{c' \in [h_i]} y_{c'} \mathbf{c}_{c'} \right). \tag{10}$$

**Proof.**  Let us apply the chain rule. For every $i \in [k]$:

$$\nabla_{\mathbf{z}_i} \mathcal{L}^{\text{NeSy}}(\mathbf{y}, \text{K}) = \sum_{c \in [h_i]} \frac{\partial \mathcal{L}^{\text{NeSy}}(\mathbf{y}, \text{K})}{\partial y_c} \nabla_{\mathbf{z}_i} y_c$$

$$\nabla_{\mathbf{z}_i} y_c = \nabla_{\mathbf{z}_i} \left( \frac{\exp(-||\mathbf{z}_i - \mathbf{c}_c||_2^2)}{\sum_{c' \in [h_i]} \exp(-||\mathbf{z}_i - \mathbf{c}_{c'}||_2^2)} \right)$$

In order to ease the calculations, we calculate the partial derivative of $y_c$ with respect to the $j$th element of the embedding $\mathbf{z}_i$, which we will refer to as $z_i^j$.

Let $d_c = -||\mathbf{z}_i - \mathbf{c}_c||_2^2$ then:

$$\frac{\partial y_c}{\partial z_i^j} = \sum_{c' \in [h_i]} \frac{\partial y_c}{\partial d_c} \frac{\partial d_c}{\partial z_i^j}$$

$$= y_c \left[ (1 - y_c) \frac{\partial d_c}{\partial z_i^j} - \sum_{c' \in [h_i], c' \neq c} y_{c'} \frac{\partial d_c}{\partial z_i^j} \right].$$

Since $\frac{\partial d_c}{\partial z_i^j} = -2(z_i^j - e_c^j)$, $e_c^j$ being the $j$th element of the vector $\mathbf{c}_c$, then

$$\frac{\partial y_c}{\partial z_i^j} = y_c \left[ (1 - y_c)(-2(z_i^j - e_c^j)) - \sum_{c' \in [h_i], c' \neq c} y_{c'} (-2(z_i^j - e_{c'}^j)) \right]$$

$$= 2 y_c \left[ -(z_i^j - e_c^j) + \sum_{c' \in [h_i]} y_{c'}(z_i^j - e_{c'}^j) \right] \qquad \text{since} \quad \sum_{c' \in [h_i]} y_{c'} = 1$$

$$= 2 y_c \left[ e_c^i - \sum_{c' \in [h_i]} y_{c'} e_{c'}^j \right]$$

We can hence conclude:

$$\nabla_{\mathbf{z}_i} \mathcal{L}^{\text{NeSy}}(\mathbf{y}, \text{K}) = 2 \sum_{c \in [h_i]} \frac{\partial \mathcal{L}^{\text{NeSy}}}{\partial y_c} y_c \left( \mathbf{c}_c - \sum_{c' \in [h_i]} y_{c'} \mathbf{c}_{c'} \right).$$

$\square$

# B  Proof Corollary 4.2

**Statement.** Let $\mathbf{y}$ be the output of a prototypical NeSy predictor reasoning over the knowledge K and the set of concepts $\mathcal{C} = [h_1] \times \ldots \times [h_k]$. If the predictor is trained using semantic loss $\mathcal{L}^{\text{NeSy}}(\mathbf{y}, \text{K})$ as defined in Equation 7, then for every $i \in [k]$ the error calculated with respect to the embedding $\mathbf{z}_i$ is equal to:

$$\nabla_{\mathbf{z}_i} \mathcal{L}^{\text{NeSy}}(\mathbf{y}, \text{K}) = 2 \sum_{c \in [h_i]} \left[ \frac{y_c - \mathbb{E}[Y_c \mid \mathbf{y}, \nu \models \text{K}]}{y_c(1 - y_c)} \right] y_c \left( \mathbf{c}_c - \sum_{c'} \mathbf{c}_{c'} y_{c'} \right). \tag{11}$$

**Proof.** Recall that the semantic loss is defined as:

$$\mathcal{L}^{\text{NeSy}}(\mathbf{y}, K) = -\log \sum_{\nu \models K} \prod_{c \in \bigcup_i [h_i]} (y_c)^{\nu(c)} (1 - y_c)^{1-\nu(c)},$$

where each $\nu : \bigcup_i [h_i] \to \{0,1\}$ represents a boolean assignment of the variables in $\bigcup_i [h_i]$. Given Theorem 4.1, to prove our statement, we need to calculate the value of $\frac{\partial \mathcal{L}^{\text{NeSy}}(\mathbf{y}, K)}{\partial y_c}$:

$$\frac{\partial \mathcal{L}^{\text{NeSy}}(\mathbf{y}, K)}{\partial y_c} = -\frac{\sum_{\nu \models K} \prod_{c' \in \bigcup_i [h_i], c' \neq c} (y_{c'})^{\nu(c')} (1 - y_{c'})^{1-\nu(c')} \frac{\partial}{\partial y_c} \left( (y_c)^{\nu(c)} (1 - y_c)^{1-\nu(c)} \right)}{\sum_{\nu \models K} \prod_{c \in \bigcup_i [h_i]} (y_c)^{\nu(c)} (1 - y_c)^{1-\nu(c)}}$$

where

$$\frac{\partial}{\partial y_c} \left( (y_c)^{\nu(c)} (1 - y_c)^{1-\nu(c)} \right) = (1 - y_c)^{1-\nu(c)} (y_c)^{\nu(c)} \left( \frac{\nu(c)}{y_c} - \frac{1 - \nu(c)}{1 - y_c} \right)$$

$$= (1 - y_c)^{1-\nu(c)} (y_c)^{\nu(c)} \left( \frac{\nu(c) - y_c}{y_c (1 - y_c)} \right).$$

Substituting we obtain:

$$\frac{\partial \mathcal{L}^{\text{NeSy}}(\mathbf{y}, K)}{\partial y_c} = -\frac{\sum_{\nu \models K} \prod_{c' \in \bigcup_i [h_i]} (y_{c'})^{\nu(c')} (1 - y_{c'})^{1-\nu(c')} \left( \frac{\nu(c) - y_c}{y_c (1 - y_c)} \right)}{\sum_{\nu \models K} \prod_{c \in \bigcup_i [h_i]} (y_c)^{\nu(c)} (1 - y_c)^{1-\nu(c)}}$$

Since each concept $c \in \bigcup_i [h_i]$ corresponds to an independent random event whose truth assignment is modelled by a Bernoulli random variable $Y_c \sim \text{Bernoulli}(y_c)$, the joint probability distribution over assignments $\nu : \bigcup_i [h_i] \to \{0,1\}$ factorizes as:

$$p(\nu \mid \mathbf{y}) = \prod_{c \in \bigcup_i [h_i]} (y_c)^{\nu(c)} (1 - y_c)^{1-\nu(c)}.$$

This allows us to compute the expected value of each $Y_c$ conditioned on the satisfaction of K as:

$$\mathbb{E}[Y_c \mid \mathbf{y}, \nu \models K] = \frac{\sum_{\nu \models K} p(\nu \mid \mathbf{y}) \cdot \nu(c)}{\sum_{\nu \models K} p(\nu \mid \mathbf{y})}.$$

We can rewrite the error calculated with respect to the output for the class $c$ as:

$$\frac{\partial \mathcal{L}^{\text{NeSy}}(\mathbf{y}, K)}{\partial y_c} = \frac{y_c - \mathbb{E}[Y_c \mid \nu \models K, \mathbf{y}]}{y_c (1 - y_c)}.$$

The statement thus follows. $\qquad \square$

## C   Proof of Proposition 5.1

**Statement**   Consider a prototypical NeSy predictor inducing $p_\theta(\mathbf{Y}|\mathbf{X}; K)$. Assume $\mathcal{G} = [h_1] \times \ldots \times [h_k]$ and let $\mathcal{H}_i \subseteq [h_i]$ be the set of concepts with at least one labelled datapoint from a dataset $\mathcal{D}$. Under [**A1,2,3**] the number of deterministic optima for $p_\theta(\mathbf{C} \mid \mathbf{G})$ is

$$\sum_{\alpha \in \mathcal{A}} \mathbb{1}\Big( \bigwedge_{\mathbf{g} \in \text{supp}(\mathbf{G})} \Big( \bigwedge_{i \in [k]} \bigwedge_{g_j \in \mathcal{H}_i} (\alpha(\mathbf{g})_i = g_i \wedge g_i = g_j) \Big) \wedge \Big( (\beta_K \circ \alpha)(\mathbf{g}) = \beta_K(\mathbf{g}) \Big) \Big),$$

$\alpha$ being the mapping induced by $p_\theta(\mathbf{C} \mid \mathbf{G})$ and $\mathcal{A}$ being the set of all possible mappings $\alpha$.

**Proof.**   We assume that:

  [**A1**] the distribution $p^*(\mathbf{X} \mid \mathbf{G}, \mathbf{S})$ is induced by a map $\phi \colon (\mathbf{g}, \mathbf{s}) \mapsto \mathbf{x}$, where $\phi$ is invertible and smooth over $\mathbf{s}$;

**[A2]** the distribution $p^*(\mathbf{Y} \mid \mathbf{G}; \mathrm{K})$ is induced by a deterministic mapping $\beta_{\mathrm{K}} : \mathbf{g} \rightarrow \mathbf{y}$ and $p^*(\mathbf{y} \mid \mathbf{g}; \mathrm{K}) = 0$, for all $(\mathbf{g}, \mathbf{y})$ not compliant with K;

**[A3]** given a bijective assignment $\sigma : \mathcal{G} \rightarrow \mathcal{G}$ such that $\sigma(\mathbf{g})_i \in [h_i]$ with $i = 1, \dots, k$, it holds that for every datapoint $\mathbf{x} = \phi(\mathbf{g}, \mathbf{s})$, $\mathrm{argmin}_{j \in [h_i]} ||f_\theta^i(\mathbf{x}) - \mathbf{c}_j||_2^2 = \sigma(\mathbf{g})_i$.

Under **[A1]**, the data distribution is induced by a smooth, invertible map $\phi : (\mathbf{g}, \mathbf{s}) \mapsto \mathbf{x}$, so every datapoint can be written as $\mathbf{x} = \phi(\mathbf{g}, \mathbf{s})$.

Under **[A1]** and **[A3]**, any deterministic predictor $p_\theta(\mathbf{C} \mid \mathbf{X})$ that is constant over the support of $p(\mathbf{X} \mid \mathbf{g})$ corresponds uniquely to a deterministic mapping $\alpha : \mathcal{G} \rightarrow \mathcal{G}$, defined by $\alpha(\mathbf{g}) = \sigma(\mathbf{g})$. Conversely, each $\alpha$ induces a deterministic distribution $p_\theta(\mathbf{C} \mid \mathbf{X})$; thus, it suffices to characterize the possible assignments $\alpha \in \mathcal{A}$ given the available supervisions in $\mathcal{D}$.

We now proceed to characterise the number of deterministic optima. For any $i \in [k]$ and supervised label $g_j \in \mathcal{H}_i$ the cross-entropy loss wrt. the concepts is maximised if and only if $p_\theta(C_i = g_j \mid \mathbf{g}) = 1$, which corresponds to a deterministic mapping $\alpha(\mathbf{g})_i = g_j$. Hence, given $i \in [k]$ for all the deterministic optima it must hold:

$$\bigwedge_{g_j \in \mathcal{H}_i} (\alpha_i(\mathbf{g}) = g_i \wedge g_i = g_j).$$

When accounting for all the $\mathcal{H}_i$ and $i \in [k]$, we obtain:

$$\bigwedge_{\mathbf{g} \in \mathrm{supp}(\mathbf{G})} \bigwedge_{i \in [k]} \bigwedge_{g_j \in \mathcal{H}_i} (\alpha_i(\mathbf{g}) = g_i \wedge g_i = g_j).$$

Under **[A2]**, a NeSy predictor achieves zero NeSy loss wrt. the final label if and only if it is compliant with K, then it must hold:

$$(\beta_{\mathrm{K}} \circ \alpha)(\mathbf{g}) = \beta_{\mathrm{K}}(\mathbf{g})$$

Hence, under **[A1,2,3]** the number of deterministic optima for $p_\theta(\mathbf{C} \mid \mathbf{G})$ is

$$\sum_{\alpha \in \mathcal{A}} \mathbb{1} \Big( \bigwedge_{\mathbf{g} \in \mathrm{supp}(\mathbf{G})} \Big( \bigwedge_{i \in [k]} \bigwedge_{g_j \in \mathcal{H}_i} (\alpha(\mathbf{g})_i = g_i \wedge g_i = g_j) \Big) \wedge \Big( (\beta_{\mathrm{K}} \circ \alpha)(\mathbf{g}) = \beta_{\mathrm{K}}(\mathbf{g}) \Big) \Big).$$

$\square$

## D    Tasks Details

The tasks belong to the `rs-bench`[2] benchmark suite. In this benchmark, all ready-made data sets and generated datasets are distributed under the CC-BY-SA 4.0 license, with the exception of `Kand-Logic`, which is derived from `Kandinsky-patterns` [34] and as such is distributed under the GPL-3.0 license. Regarding the code, most of the code in the repo is distributed under the BSD 3 license, with the exception of `Kand-Logic`, which is derived from `Kandinsky-patterns` and as such is distributed under the GPL-3.0 license.

`MNIST-EvenOdd` [28, 30]   The dataset presents a more complex version of the standard `MNIST-Addition` task. Each input is a pair of handwritten $28 \times 28$ digits labelled with their sum (background knowledge). The goal is to correctly classify the digits (concepts) while predicting their sum (final label). Notably, the restricted support is designed to stir up shortcut reasoning, with a total of 49 distinct RSs a model may take. Since the training support is incomplete, thus shortcuts cannot be avoided by simply processing each digit in isolation. During training only 16 possible pairs of digits out of 100 are given - 8 comprising only even digits and 8 only odd digits - while at test time the model is presented with out-of-distribution pairs. The list of all possible training combinations and their labels are given in Table 6. Overall, the training set contains 6720 data, the validation set 1920, and the test set 960.

`Kand-Logic` [34, 30]   Inspired by the artworks of Wassily Kandinsky, this task involves non-trivial perception to classify shapes and colours, and logical reasoning to infer

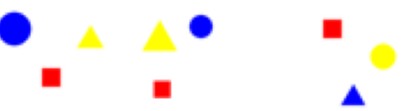

Figure 8: Datapoint from `Kand-Logic`.

[2]Link: https://github.com/unitn-sml/rsbench-code

Table 6: Pairs of digits and their label available in `MNIST-EvenOdd`.

| Example Pair | Label | Example Pair | Label |
|---|---|---|---|
| (0, 6) | 6 | (1, 5) | 6 |
| (2, 8) | 10 | (3, 7) | 10 |
| (4, 6) | 10 | (1, 9) | 10 |
| (4, 8) | 12 | (3, 9) | 12 |

patterns among them. Each datapoint $(\mathbf{x}_1, \mathbf{x}_2, \mathbf{x}_3)$ consists of three $64 \times 64$ images, representing three geometric primitives each. An example is depicted in Figure 8. Every sample is labelled as 1 whether *all* images either include the *same* number of primitives with the *same* shape or with the *same* colour, and 0 otherwise. In order to formally specify the background knowledge K, we define six binary classes, deterministically determined from the concept classes:

- `SameS`$_i$ (resp. `SameC`$_i$) capturing whether all the primitives in $\mathbf{x}_i$ have the same shape (resp. color) or not,

- `DiffS`$_i$ (resp. `DiffC`$_i$) capturing whether all primitives in $\mathbf{x}_i$ have different shape (resp. color) or not, and

- `TwoS`$_i$ (resp. `TwoC`$_i$) capturing whether exactly two of the three primitives in $\mathbf{x}_i$ have the same shape (resp. color) or not.

The background knowledge is given by

$$
\begin{aligned}
\text{K} \iff &((\texttt{DiffS}_1 \wedge \texttt{DiffS}_2 \wedge \texttt{DiffS}_3) \vee (\texttt{TwoS}_1 \wedge \texttt{TwoS}_2 \wedge \texttt{TwoS}_3) \vee \\
&(\texttt{SameS}_1 \wedge \texttt{SameS}_2 \wedge \texttt{SameS}_3) \vee (\texttt{DiffC}_1 \wedge \texttt{DiffC}_2 \wedge \texttt{DiffC}_3) \vee \\
&(\texttt{TwoC}_1 \wedge \texttt{TwoC}_2 \wedge \texttt{TwoC}_3) \vee (\texttt{SameC}_1 \wedge \texttt{SameC}_2 \wedge \texttt{SameC}_3)).
\end{aligned}
$$

Thus specifying that all images either include the same number of primitives with the same shape or with the same colour. For example, Figure 8 shows an image whose label is 1 because all the primitives appearing in each image have different shapes and different colours. Overall, the training set consists of 4000 datapoints, while the validation and test sets of 1000. Each image in turn shows three shapes, for a total of 9 primitives per datapoint.

`BDD-OIA` [50]   This real-world and high-stake autonomous driving dataset was originally proposed as an extension of BDD-100k [52] and contains a total of ∼22.5k frames captured from the ego-vehicle. Each frame, whose size is $720 \times 1280$, receives labels from {*MoveForward*, *Stop*, *TurnLeft*, *TurnRight*} to reflect the right action to be undertaken. The goal is to correctly classify the reasons (concepts) behind each action (final label, e.g. given *Person* $\vee$ *RedLight* $\implies$ *Stop*, *Person* caused *Stop*). Frames are processed using a Faster-RCNN model [38] and the first pre-trained layer from [39] so as to obtain 2048-dimensional embeddings expressing the features present in each training scene. These embeddings are then used to train the NeSy models. Each frame (or equivalently, embedding) is annotated with one of 4 possible labels {*MoveForward*, *Stop*, *TurnLeft*, *TurnRight*}, each indicating a possible action taken by the ego-vehicle. The learning task, differently from the previous ones, is multi-label: multiple different actions can be labelled as possible in the provided scene with the only exception of *MoveForward* and *Stop* that are clearly mutually exclusive choices. Additionally, each embedding is annotated with a subset of the 21 concepts that might cause the action:

{*GreenLight*, *RedLight*, *TrafficSign*, *Follow*, *RoadClear*, *Car*, *Person*, *Rider*, *OtherObstacle*, *LeftLane*, *LeftGreenLight*, *LeftFollow*, *NoLeftLane*, *LeftObstacle*, *LeftSolidLine*, *RightLane*, *RightGreenLight*, *RightFollow*, *NoRightLane*, *RightObstacle*, *RightSolidLine*}.

The rules in K specify concept-to-concept and concept-to-action relations. For example:

$$
\begin{aligned}
&RedLight \rightarrow \neg GreenLight, \\
&\neg Car \vee Person \vee Rider \vee OtherObstacle \leftrightarrow RoadClear, \\
&GreenLight \vee Follow \vee RoadClear \rightarrow MoveForward.
\end{aligned}
$$

For the complete specification of the background knowledge K see Appendix C.2.5 of [29].

This dataset presents us with several challenges:

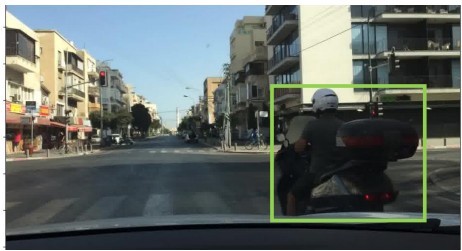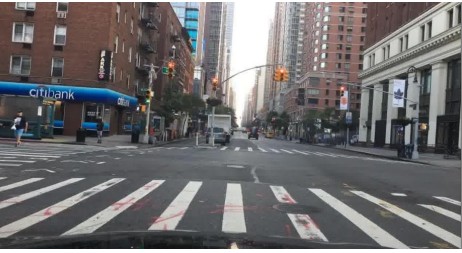

(a) Rider in the scene              (b) Rider no longer in the scene

Figure 9: Both Images in the `BDD-OIA` dataset are labelled as *Rider* inducing a *Stop* action.

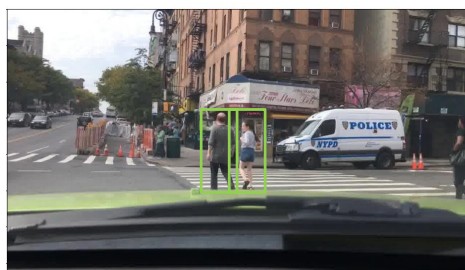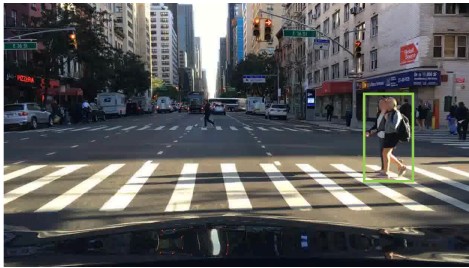

(a) Pedestrian caused the ego-vehicle to stop    (b) Pedestrian did not cause the ego-vehicle to stop

Figure 10: Both Images in the `BDD-OIA` show instances of the *Pedestrian* class, yet only (a) is labelled as such since the pedestrians in (b) did not cause the ego-vehicle to halt.

- Many concepts are under-represented, making it difficult for the model to learn to recognize them effectively. In particular the exact distribution of the concepts is shown in Table 7. As we can see the training data are heavily unbalanced, with some concepts being represented by only a few examples;
- The labels are noisy, and may contain errors or inconsistencies. For example some image may be labelled with both the actions *MoveForward* and *Stop*, which are mutually exclusive. This is typical of large datasets, especially the real world ones like BDD-OIA, where the labelling process is often done by humans and may be subject to errors or inconsistencies;
- In some frames temporal information may be lost. See Figure 9 for an example where the temporal information is needed to correctly label the concepts. Indeed, in the Figure we can see that both the frames are labelled as *Rider*, but in (a) it is clear that *Rider* is present and cause for the stop, while in (b) the *Rider* is already far away and we are able to infer the correct concept only by looking at the previous ones; conversely, the embeddings of the scene are generated by looking at a single frame at a time. This limitation is known, as it was already highlighted by [39];
- Single frames may lack contextual information, which can be important for understanding the scene. For example, in Figure 10 we can see that the context of the pedestrians is crucial for understanding the correct action to be taken. Indeed, even though in both frames the pedestrians are crossing the street, the context is different: in (a) the pedestrians are crossing the street and they are the *only* reason for the stop, while in (b) there is also a traffic light that is red. This means that in (a) the pedestrians are labelled as present because the car stops in front of them, while in (b) the pedestrians are not labelled as present because the car stops due to the red light.

## E   Experimental Setup

**Neural Architectures**    As backbone concept extractors for the baseline models we used the architecture [5]. On the other hand, as a neural prototypical extractor model for the `MNIST-Even-Odd` task, we used the same convolutional neural network as in [42] composed of four convolutional blocks: the first one is a $1 \times 64$ and the others $64 \times 64$, respectively. Each blocks consists of a $3 \times 3$

convolutional filter, a batch normalization layer [20], a ReLU nonlinearity, and a $2 \times 2$ max-pooling layer. One such prototypical network is used to classify all digits. The same architecture has also been employed for the `Kand-Logic` task, with the exception that prototypical networks work on three RGB channels and their inputs are primitives detected in their corresponding images using YOLO [37]. In particular, we adopted YOLOv11 [21] that exhibits enhanced detection capabilities in virtue of the introduction of cross-stage partial blocks, spatial pyramid pooling, and convolutional blocks with Parallel Spatial Attention components.

For running our experiments on `Kand-Logic` the nano version of YOLO was sufficient for the purpose of primitive detection. The model was trained on the same support set constructed through data augmentation for prototypical networks. We employed two prototypical extractors, sharing the same architecture and support set, for the task of classifying primitives shapes and colours. For this tasks, the backbone extractors for baseline are modified so as to process independently each primitive extracted using YOLO. We also employ two distinct concept extractors to process in a disentangled manner both shapes and colours, matching exactly the same setup for prototypical models.

As for `BDD-OIA`, each original frame is processed following [39] by the means of a Faster-RCNN model that was pre-trained on MS-COCO and then fine-tuned on BDD-100k [52]. Through the Faster-RCNN backbone the features for the objects in the scene are computed (e.g., traffic lights). Subsequently, we employ the first pre-trained layer from [39] to produce a 2048-dimensional which we use to train the NeSy models.

To process the embeddings we created, the standard architecture of prototypical extractors from [42] was adapted to work with 1D convolutions. More in details, the prototypical architecture we employed adopts three convolutional blocks, including a $1 \times 32$, $32 \times 64$ and $64 \times 128$ layer respectively, each comprising 5-wide convolutional filters, batch normalization, ReLU activation and a 1D kernel size 2 max-pooling layer. The last block replaces max-pooling with adaptive average pooling of length 1. Finally, the output is flattened and further processed through a linear layer, 1D batch normalization and ReLU to yield a 256-dimensional embedding. For `BDD-OIA`, 21 prototypical extractors were employed, each classifying one concepts. As for the case of `Kand-Logic`, we modify the standard baseline to use the same number of backbones as the prototypical models.

All of our models were trained via SGD with Adam [22], a learning rate equal to $10^{-3}$, and weight decay as the regularization method.

**Support sets**    For most of the experiments we worked with **at most one** annotation per concept. As we showed in Section 6 RQ1, this is enough to curb the reasoning shortcut problem. Under this premise, in many experiments we decided to use standard data augmentation techniques on the annotated data from the support set to pretrain and supervise standard NeSy models (as shown in Section 6) and/or slightly improve the generalization ability of our prototypical extractors. For example, In `MNIST-EvenOdd`, each annotated concept gets augmented through rotations (40 data points generated), translation (144 data points), scaling (10 data points), elastic distortion (25 data points) and Gaussian noise addition (4 data points). These transformations try to mimic some of the variability observed in handwriting.

Likewise, for `Kand-Logic` the chosen transformations are scaling, anti-aliasing addition and translation, as the generated data are representative of the primitives the actual dataset contains. In the latter case, we generate 163 data points per (shape, colour) pair. For `BDD-OIA`, given a concept $j$, at least 6 samples are drawn with $i$ label 1. Then positive (resp. negative) samples are added from samples positive for each class $j'$, $j \neq j'$, where $j$ also occurs (resp. does not occur). We end up with $\approx 200$ samples per class, more than 85% of which are negatives. No data augmentation is used and the prototypical networks are all trained using one training batch randomly sampled at training time, with some classes being significantly under-represented, as shown in Table 7.

**Hyperparameters**    All the experiments were run across ten seeds: 0, 128, 256, 512, 1024, 2048, 4096, 8192, 16384, 32768. The value of the following hyperparameters is used consistently for all the experiments when training the NeSy models to predict weak labels for each example in $\mathcal{D}$:

- exponential decay rate $\beta$: 0.99;
- number of epochs: 10 for `MNIST-EvenOdd` and `Kand-Logic`, 40 for `BDD-OIA`;
- semantic loss weight $w_{\mathrm{sl}}$: 10.0;
- warm-up steps for learning rate increase: 0.

Table 7: Concept-level number and percentage of occurrences for each class in 16080 training frames from the `BDDOIA` dataset.

| Concept | Num. of occurrences | Percentage |
|---|---|---|
| Green Light | 5478 | 34.07% |
| Follow Traffic | 2448 | 15.22% |
| Road is clear | 3422 | 21.28% |
| Red Light | 3766 | 23.42% |
| Traffic Sign | 1095 | 6.81% |
| Obstacle Car | 171 | 1.06% |
| Obstacle Person | 116 | 0.72% |
| Obstacle Rider | 3703 | 23.03% |
| Obstacle Other | 329 | 2.05% |
| No Lane Left | 100 | 0.62% |
| Obstacle on Left | 464 | 2.89% |
| Solid Line Left | 220 | 1.37% |
| On Right Turn Lane | 115 | 0.72% |
| Traffic Light Allows Right | 629 | 3.91% |
| Front Car Turning Right | 270 | 1.68% |
| No Lane Right | 3182 | 19.79% |
| Obstacle on Right | 3177 | 19.76% |
| Solid Line Right | 2587 | 16.09% |
| On Left Turn Lane | 4303 | 26.76% |
| Traffic Light Allows Left | 2796 | 17.39% |
| Front Car Turning Left | 1563 | 9.72% |

On the other hand, these hyperparameters were used across different models and tasks for the episodic training of prototypical networks:

- learning rate for prototypical networks training: $10^{-3}$;
- number of support and query examples for prototypical networks training: 10 (5 each) if the support set is expanded with data augmentation and 2 (1 each) otherwise;
- number of episodes per training epoch: 100.

As for the model and application specific parameters, for every task we report a detailed list of all the hyperparameters we used for training NeSy models over the weakly labelled examples. We use $B$ for batch size, $\gamma$ for the learning rate and $\omega$ for the weight decay. For LTN $p$ refers to the hyperparameter for quantifiers. When the models are trained with Shannon Entropy, the parameter $w_h$ indicates its weight. Lastly, $m$ denotes the embedding dimension for the prototypes extracted.

`MNIST-EvenOdd`.

- DPL: $B = 32$, $\gamma = 10^{-3}$, $\omega = 10^{-4}$, $w_h = 1$, $m = 64$,
- LTN: $B = 64$, $\gamma = 10^{-3}$, $\omega = 10^{-4}$, $p = 6$, $w_h = 10$, `AND` operation uses Godel, `OR` and `IMPLICATION` Product as fuzzy semantics, $m = 64$,
- SL: $B = 32$, $\gamma = 10^{-3}$, $\omega = 10^{-4}$, $m = 64$.
- CCN+: $B = 32$, $\gamma = 10^{-3}$, $\omega = 10^{-4}$, $m = 64$.
- ABL: $B = 32$, $\gamma = 10^{-3}$, $\omega = 10^{-4}$

`Kand-Logic`.

- DPL: $B = 32$, $\gamma = 10^{-4}$, $\omega = 0$, $w_h = 1$, $m = 1024$,
- LTN: $B = 64$, $\gamma = 10^{-3}$, $\omega = 10^{-3}$, $p = 8$, $w_h = 0.8$, `AND` operation uses Godel, `OR` and `IMPLICATION` Product as fuzzy semantics, $m = 1024$,
- SL: $B = 32$, $\gamma = 10^{-3}$, $\omega = 10^{-4}$, $m = 1024$.

`BDD-OIA`.

- DPL: $B = 128$, $\gamma = 10^{-3}$, $\omega = 10^{-4}$, $w_h = 1$, $m = 256$.

Notice that lowering the value and $m$ in the case of `BDD-OIA` was necessary as we observed the prototypical networks can in some cases easily overfit the training embeddings.

**Computing Infrastructure**   All the experiments run on a machine equipped with the following:

- Intel® Xeon® CPU E5-2620 v3 @ 2.40GHz, 24 Cores, 48 Threads;
- 4 NVidia TITAN Xp GPU, 12 GB GDDR5X, 3840 CUDA Cores.

whose operating system is Ubuntu 20.04.5 LTS and CUDA Version 12.2. For realizing the code of our experiments we used Python 3.20 and Pytorch 1.13.0.

# F   Additional Results

In this Section we include the additional results we obtained for each research question and experiment which we could not include in the main body of the paper due to space constraints.

**RQ1:**   *Can Prototypical Neurosymbolic AI avoid reasoning shortcuts?*   The complete results for prototypical NeSy models are reported in Table 8,  10 and 9 for the `MNIST-EvenOdd`, `Kand-Logic` and `BDD-OIA` task respectively. In the Tables our mitigation strategy resorting on prototypical concept extractor is denoted as *PNet*, while *PNet★* is the variant who employs standard data augmentation techniques to enhance the variety of the support set, as explained in Appendix E, on `MNIST-EvenOdd` and `Kand-Logic`. Regarding `MNIST-EvenOdd` we introduced several additional baselines exploiting the annotations from the prototypical support sets in various ways. In particular, the other considered mitigations are:

- *support set $S$ concept supervision*: similarly to the mitigation strategy C, proposed in [29], during training we supervise the NeSy models with the annotations in $S$ that undergone an additional data augmentation step, so as to steer them towards the correct predictions. We denote this variant as *C(S)★*;
- *pretraining*: the models' backbones are trained in a supervised fashion on the support set before training them on the target task. In this way the baselines can leverage the support set information to learn better initial parameters. The main text discusses pretraining (*Pre★*) on the support set until convergence before training on the weakly supervised data. We also tested epoch pretraining (*EPre★*), where the model is pretrained on the support set for one epoch before each training epoch on the actual task. This second pretraining strategy is designed to closely resemble the execution of Algorithm 1 used to train our prototypical models;
- combinations of pretraining and support set concept supervision.

These additional baselines are useful to demonstrate that the performance gains we present are due to our architecture, rather than the support set information. From Table 8 two main observations can be made. First, Prototypical Neurosymbolic models effectively avoid reasoning shortcuts. Second, none of the baselines introduced is competitive with their prototypical counterpart across *all* metrics and *all* the models studied.

Regarding this second point, observe for instance that *EPre★* performs competitively with our solution on Semantic Loss, but it is outperformed by it on DeepProbLog and even more so on Logic Tensor Networks.

Interestingly, Prototypical Neurosymbolic models also maintain - often even improve - their performance on label scores compared to the baselines affected by the reasoning shortcuts. This shows that enhancing concept alignment is also greatly beneficial to the overall task performance. The only exception to the above concerns the F1(Y) score. This metric is lower than that of their respective baselines in the case of Logic Tensor Networks and Semantic Loss. Conversely we do not observe this drop in performance for DeepProbLog, where the F1(Y) score is actually improved wrt. most baselines. The difference can be explained by the different ways in which these models integrate the reasoning and learning components. Indeed, DeepProbLog leverages constrained-output marginalization, and it allocates probability mass only to outputs that satisfy the logical constraints: as a consequence, during training gradient updates are considered only for digits participating in consistent sum predictions. In contrast, the semantic objective of Logic Tensor Networks and Semantic Loss allows for occasional miss-classification.

Finally, notice that the use of data augmentation plays only a marginal role in avoiding reasoning shortcuts in Prototypical Neurosymbolic models: the difference in both the concept and label scores

Table 8: Different mitigations strategies compared across NeSy models, trained with 100% of `MNIST-EvenOdd` data. Best results are in bold and second best underlined.

| Model / Mitigation | Acc(C)($\uparrow$) | F1(C)($\uparrow$) | Acc(Y)($\uparrow$) | F1(Y)($\uparrow$) | Cls(C)($\downarrow$) |
|---|---|---|---|---|---|
| **Logic Tensor Network** | | | | | |
| *Baseline* | $0.13_{\pm 0.07}$ | $0.05_{\pm 0.04}$ | $0.61_{\pm 0.18}$ | $0.66_{\pm 0.10}$ | $0.79_{\pm 0.08}$ |
| *R* | $0.14_{\pm 0.07}$ | $0.05_{\pm 0.02}$ | $0.72_{\pm 0.00}$ | $0.71_{\pm 0.01}$ | $0.70_{\pm 0.00}$ |
| *H* | $0.55_{\pm 0.14}$ | $0.55_{\pm 0.14}$ | $0.77_{\pm 0.09}$ | $0.85_{\pm 0.08}$ | $0.08_{\pm 0.02}$ |
| *C* | $0.26_{\pm 0.00}$ | $0.12_{\pm 0.00}$ | $0.77_{\pm 0.00}$ | $0.78_{\pm 0.00}$ | $0.71_{\pm 0.00}$ |
| *C(S)$^\star$* | $0.46_{\pm 0.31}$ | $0.37_{\pm 0.35}$ | $0.80_{\pm 0.12}$ | $0.61_{\pm 0.10}$ | $0.49_{\pm 0.29}$ |
| *R+H* | $0.65_{\pm 0.10}$ | $0.61_{\pm 0.10}$ | $0.74_{\pm 0.20}$ | $0.81_{\pm 0.20}$ | $0.21_{\pm 0.16}$ |
| *R+C* | $0.23_{\pm 0.00}$ | $0.10_{\pm 0.00}$ | $0.75_{\pm 0.00}$ | $0.75_{\pm 0.00}$ | $0.70_{\pm 0.00}$ |
| *H+C* | $\mathbf{0.98}_{\pm 0.00}$ | $\mathbf{0.98}_{\pm 0.00}$ | $\mathbf{0.97}_{\pm 0.00}$ | $\mathbf{0.98}_{\pm 0.00}$ | $\underline{0.01}_{\pm 0.00}$ |
| *R+H+C* | $\underline{0.95}_{\pm 0.00}$ | $0.95_{\pm 0.00}$ | $0.91_{\pm 0.00}$ | $\underline{0.94}_{\pm 0.00}$ | $0.04_{\pm 0.00}$ |
| *Pre$^\star$* | $0.51_{\pm 0.22}$ | $0.39_{\pm 0.25}$ | $0.89_{\pm 0.07}$ | $0.70_{\pm 0.06}$ | $0.45_{\pm 0.23}$ |
| *Pre$^\star$+C(S)$^\star$* | $0.72_{\pm 0.27}$ | $0.64_{\pm 0.34}$ | $0.94_{\pm 0.06}$ | $0.71_{\pm 0.05}$ | $0.24_{\pm 0.25}$ |
| *Pre$^\star$* | $0.27_{\pm 0.24}$ | $0.19_{\pm 0.26}$ | $0.78_{\pm 0.09}$ | $0.58_{\pm 0.08}$ | $0.62_{\pm 0.21}$ |
| *PNet (ours)* | $\underline{0.95}_{\pm 0.03}$ | $0.95_{\pm 0.03}$ | $0.91_{\pm 0.05}$ | $0.71_{\pm 0.03}$ | $\mathbf{0.00}_{\pm 0.00}$ |
| *PNet$^\star$ (ours)* | $\mathbf{0.98}_{\pm 0.00}$ | $\underline{0.97}_{\pm 0.00}$ | $\underline{0.95}_{\pm 0.01}$ | $0.73_{\pm 0.00}$ | $\mathbf{0.00}_{\pm 0.00}$ |
| **Semantic Loss** | | | | | |
| *Baseline* | $0.08_{\pm 0.12}$ | $0.04_{\pm 0.04}$ | $\mathbf{0.99}_{\pm 0.00}$ | $\mathbf{0.99}_{\pm 0.00}$ | $0.68_{\pm 0.04}$ |
| *R* | $0.01_{\pm 0.00}$ | $0.01_{\pm 0.00}$ | $\mathbf{0.99}_{\pm 0.00}$ | $\mathbf{0.99}_{\pm 0.00}$ | $0.70_{\pm 0.00}$ |
| *H* | $0.01_{\pm 0.00}$ | $0.01_{\pm 0.00}$ | $\mathbf{0.99}_{\pm 0.00}$ | $\mathbf{0.99}_{\pm 0.00}$ | $0.70_{\pm 0.00}$ |
| *C* | $0.30_{\pm 0.01}$ | $0.22_{\pm 0.01}$ | $\mathbf{0.99}_{\pm 0.00}$ | $\mathbf{0.99}_{\pm 0.00}$ | $0.60_{\pm 0.01}$ |
| *C(S)$^\star$* | $0.56_{\pm 0.24}$ | $0.45_{\pm 0.27}$ | $\mathbf{0.99}_{\pm 0.01}$ | $\mathbf{0.99}_{\pm 0.01}$ | $0.26_{\pm 0.24}$ |
| *R+H* | $0.01_{\pm 0.00}$ | $0.01_{\pm 0.00}$ | $\mathbf{0.99}_{\pm 0.00}$ | $\mathbf{0.99}_{\pm 0.00}$ | $0.70_{\pm 0.00}$ |
| *R+C* | $0.54_{\pm 0.07}$ | $0.46_{\pm 0.07}$ | $\mathbf{0.99}_{\pm 0.00}$ | $\mathbf{0.99}_{\pm 0.00}$ | $0.47_{\pm 0.00}$ |
| *H+C* | $0.38_{\pm 0.08}$ | $0.31_{\pm 0.07}$ | $\mathbf{0.99}_{\pm 0.00}$ | $\mathbf{0.99}_{\pm 0.00}$ | $0.50_{\pm 0.00}$ |
| *R+H+C* | $0.40_{\pm 0.10}$ | $0.35_{\pm 0.10}$ | $\mathbf{0.99}_{\pm 0.00}$ | $\mathbf{0.99}_{\pm 0.00}$ | $0.50_{\pm 0.01}$ |
| *Pre$^\star$* | $0.52_{\pm 0.25}$ | $0.42_{\pm 0.28}$ | $0.98_{\pm 0.00}$ | $0.98_{\pm 0.01}$ | $0.40_{\pm 0.24}$ |
| *Pre$^\star$+C(S)$^\star$* | $0.78_{\pm 0.22}$ | $0.71_{\pm 0.29}$ | $\mathbf{0.99}_{\pm 0.00}$ | $\mathbf{0.99}_{\pm 0.00}$ | $0.17_{\pm 0.19}$ |
| *Pre$^\star$* | $0.95_{\pm 0.10}$ | $\underline{0.94}_{\pm 0.12}$ | $\underline{0.98}_{\pm 0.00}$ | $\underline{0.98}_{\pm 0.00}$ | $\underline{0.03}_{\pm 0.09}$ |
| *PNet (ours)* | $\underline{0.98}_{\pm 0.01}$ | $\mathbf{0.98}_{\pm 0.01}$ | $0.97_{\pm 0.01}$ | $0.74_{\pm 0.00}$ | $\mathbf{0.00}_{\pm 0.00}$ |
| *PNet$^\star$ (ours)* | $\mathbf{0.99}_{\pm 0.00}$ | $\mathbf{0.98}_{\pm 0.01}$ | $0.97_{\pm 0.01}$ | $0.74_{\pm 0.00}$ | $\mathbf{0.00}_{\pm 0.00}$ |
| **DeepProbLog** | | | | | |
| *Baseline* | $0.08_{\pm 0.10}$ | $0.04_{\pm 0.05}$ | $0.87_{\pm 0.06}$ | $0.94_{\pm 0.04}$ | $0.67_{\pm 0.05}$ |
| *R* | $0.17_{\pm 0.00}$ | $0.07_{\pm 0.00}$ | $0.75_{\pm 0.00}$ | $0.74_{\pm 0.00}$ | $0.70_{\pm 0.00}$ |
| *H* | $0.01_{\pm 0.00}$ | $0.01_{\pm 0.00}$ | $\mathbf{0.99}_{\pm 0.00}$ | $\mathbf{0.99}_{\pm 0.00}$ | $0.70_{\pm 0.00}$ |
| *C* | $0.01_{\pm 0.00}$ | $0.01_{\pm 0.00}$ | $0.75_{\pm 0.00}$ | $0.84_{\pm 0.00}$ | $0.70_{\pm 0.00}$ |
| *C(S)$^\star$* | $0.26_{\pm 0.11}$ | $0.15_{\pm 0.07}$ | $0.84_{\pm 0.07}$ | $0.84_{\pm 0.08}$ | $0.68_{\pm 0.06}$ |
| *R+H* | $0.16_{\pm 0.00}$ | $0.09_{\pm 0.02}$ | $0.83_{\pm 0.06}$ | $0.85_{\pm 0.06}$ | $0.63_{\pm 0.00}$ |
| *R+C* | $0.01_{\pm 0.00}$ | $0.01_{\pm 0.00}$ | $0.75_{\pm 0.00}$ | $0.84_{\pm 0.00}$ | $0.70_{\pm 0.00}$ |
| *H+C* | $\underline{0.97}_{\pm 0.05}$ | $\underline{0.96}_{\pm 0.06}$ | $0.96_{\pm 0.02}$ | $0.97_{\pm 0.02}$ | $0.03_{\pm 0.05}$ |
| *R+H+C* | $\mathbf{0.98}_{\pm 0.00}$ | $\mathbf{0.98}_{\pm 0.00}$ | $0.95_{\pm 0.00}$ | $0.97_{\pm 0.00}$ | $\underline{0.02}_{\pm 0.00}$ |
| *Pre$^\star$* | $0.64_{\pm 0.22}$ | $0.55_{\pm 0.28}$ | $0.95_{\pm 0.06}$ | $0.95_{\pm 0.06}$ | $0.32_{\pm 0.24}$ |
| *Pre$^\star$+C(S)$^\star$* | $0.86_{\pm 0.20}$ | $0.81_{\pm 0.26}$ | $\mathbf{0.99}_{\pm 0.01}$ | $\mathbf{0.99}_{\pm 0.01}$ | $0.11_{\pm 0.18}$ |
| *Pre$^\star$* | $0.84_{\pm 0.28}$ | $0.81_{\pm 0.33}$ | $0.95_{\pm 0.07}$ | $0.95_{\pm 0.08}$ | $0.14_{\pm 0.28}$ |
| *PNet (ours)* | $\underline{0.97}_{\pm 0.03}$ | $\underline{0.96}_{\pm 0.04}$ | $\underline{0.98}_{\pm 0.02}$ | $\underline{0.98}_{\pm 0.02}$ | $\mathbf{0.00}_{\pm 0.00}$ |
| **PNet$^\star$(ours)** | $\mathbf{0.98}_{\pm 0.00}$ | $\mathbf{0.98}_{\pm 0.00}$ | $\mathbf{0.99}_{\pm 0.00}$ | $\mathbf{0.99}_{\pm 0.00}$ | $\mathbf{0.00}_{\pm 0.00}$ |

Table 9: Results of different RSs mitigation strategies for models trained on the `BDD-OIA` task employing 100% of the unsupervised data. Best results are in bold and second best are underlined.

| Model / Mitigation | Acc(C)($\uparrow$) | F1(C)($\uparrow$) | Acc(Y)($\uparrow$) | F1(Y)($\uparrow$) | Cls(C)($\downarrow$) |
|---|---|---|---|---|---|
| **DeepProbLog** | | | | | |
| $Baseline_{\texttt{inD}}$ | $0.64\pm0.04$ | $0.08\pm0.14$ | $\underline{0.76}\pm0.01$ | $\underline{0.58}\pm0.08$ | $0.81\pm0.06$ |
| $Baseline_{\texttt{inD,prD}}$ | $0.62\pm0.07$ | $0.08\pm0.13$ | $\mathbf{0.77}\pm0.01$ | $\mathbf{0.61}\pm0.04$ | $0.82\pm0.13$ |
| $C(S)^{\star}_{\texttt{inD}}$ | $\mathbf{0.83}\pm0.01$ | $0.07\pm0.16$ | $0.74\pm0.02$ | $0.46\pm0.05$ | $0.70\pm0.06$ |
| $C(S)^{\star}_{\texttt{inD,prD}}$ | $0.81\pm0.01$ | $\underline{0.11}\pm0.17$ | $\underline{0.76}\pm0.01$ | $0.53\pm0.05$ | $\underline{0.51}\pm0.15$ |
| $Pre^{\star}_{\texttt{inD}}$ | $\mathbf{0.83}\pm0.01$ | $0.05\pm0.14$ | $0.72\pm0.03$ | $0.43\pm0.05$ | $0.86\pm0.04$ |
| $Pre^{\star}_{\texttt{inD,prD}}$ | $\underline{0.82}\pm0.03$ | $0.05\pm0.13$ | $0.71\pm0.02$ | $0.52\pm0.07$ | $0.85\pm0.04$ |
| $Pre^{\star}+C(S)^{\star}_{\texttt{inD}}$ | $\mathbf{0.83}\pm0.01$ | $0.07\pm0.16$ | $0.73\pm0.02$ | $0.41\pm0.04$ | $0.75\pm0.08$ |
| $Pre^{\star}+C(S)^{\star}_{\texttt{inD,prD}}$ | $0.81\pm0.02$ | $0.10\pm0.16$ | $\underline{0.76}\pm0.05$ | $0.51\pm0.04$ | $0.52\pm0.11$ |
| $EPre_{\texttt{inD}}$ | $\mathbf{0.83}\pm0.02$ | $0.06\pm0.15$ | $0.71\pm0.03$ | $0.44\pm0.05$ | $0.82\pm0.07$ |
| $EPre_{\texttt{inD,prD}}$ | $\mathbf{0.83}\pm0.01$ | $0.05\pm0.13$ | $0.73\pm0.03$ | $0.46\pm0.10$ | $0.82\pm0.07$ |
| $PNet$ | $0.60\pm0.03$ | $\mathbf{0.16}\pm0.15$ | $0.63\pm0.05$ | $0.41\pm0.08$ | $\mathbf{0.35}\pm0.11$ |

for the prototypical models with and without data augmentation is minimal. Thus data augmentation is considered only as a complementary technique to slightly improve concept alignment.

We tested as well a more recent, state-of-the-art neurosymbolic model, namely *Coherent by Construction* NNs ($CCN^{+}$). The results are shown in Table 11 and notice they are similar to the ones we previously discussed in Table 8: this evidence confirms that integrating NeSy models with prototypes is greatly beneficial in avoiding reasoning shortcuts, regardless of the particular model considered.

To further expand our study, we then tested for shortcut reasoning the *abductive learning* (ABL) paradigm [6] as well. The results in Table 12 show that pretraining the ABL model and supervising with the augmented support set at training time ($Pre^{\star}+C(S)^{\star}$) allows to successfully reconstruct the unlabelled concepts. However, in learning the concepts the model is not synergistically aligned with the final task (sum prediction), as pointed out by the significant drop in F1(Y) score. Conversely, prototypical NeSy models can reconstruct the unlabelled digits, *while* predicting the right final label.

We run similar experiments for the `Kand-Logic` task. Here we allow for other two type of baselines:

- *Input Disentanglement* (`inD`): the architecture of the baseline processes one single concept in isolation from the rest.
- *Predicate Disentanglement* (`prD`): the architecture of the baselines processes one single concept with separate branches for each set of mutually exclusive concept classes.

Since each concept is assigned a pair of different classes (i.e., a class of the shape and colour), distinguishing between `inD` and `prD` allows to effectively isolate the contribute of disentanglement from the one of prototypes. The results in Table 10 confirm the enhanced ability of PNet models in successfully retrieving the intended semantics for the intermediate concepts, while using them so comply with background knowledge. Moreover, they indicate that combining both forms of disentanglement with pretraining is generally the second best mitigation strategy across the board.

Table 10: Different mitigations strategies compared across NeSy models, trained with 100% of `Kand-Logic` data. Best results are in bold and second best underlined.

| Model / Mitigation | Acc(C)($\uparrow$) | F1(C)($\uparrow$) | Acc(Y)($\uparrow$) | F1(Y)($\uparrow$) | Cls(C)($\downarrow$) |
|---|---|---|---|---|---|
| **Logic Tensor Network** | | | | | |
| $Baseline_{inD}$ | $0.37_{\pm0.04}$ | $0.32_{\pm0.06}$ | $0.50_{\pm0.01}$ | $0.36_{\pm0.03}$ | $0.26_{\pm0.22}$ |
| $Baseline_{inD,prD}$ | $0.36_{\pm0.06}$ | $0.22_{\pm0.06}$ | $0.50_{\pm0.01}$ | $0.34_{\pm0.02}$ | $0.64_{\pm0.16}$ |
| $C(S)^\star_{inD}$ | $0.39_{\pm0.03}$ | $0.34_{\pm0.04}$ | $0.51_{\pm0.01}$ | $0.41_{\pm0.06}$ | $0.26_{\pm0.16}$ |
| $C(S)^\star_{inD,prD}$ | $0.60_{\pm0.06}$ | $0.52_{\pm0.06}$ | $0.51_{\pm0.01}$ | $0.36_{\pm0.04}$ | $0.54_{\pm0.09}$ |
| $Pre^\star_{inD}$ | $0.57_{\pm0.01}$ | $0.56_{\pm0.01}$ | $0.51_{\pm0.00}$ | $0.43_{\pm0.01}$ | $\mathbf{0.00}_{\pm0.00}$ |
| $Pre^\star_{inD,prD}$ | $0.90_{\pm0.01}$ | $\underline{0.90}_{\pm0.01}$ | $0.76_{\pm0.03}$ | $0.74_{\pm0.04}$ | $\mathbf{0.00}_{\pm0.00}$ |
| $Pre^\star+C(S)^\star_{inD}$ | $0.43_{\pm0.02}$ | $0.40_{\pm0.02}$ | $0.50_{\pm0.00}$ | $0.34_{\pm0.01}$ | $\underline{0.16}_{\pm0.05}$ |
| $Pre^\star+C(S)^\star_{inD,prD}$ | $0.88_{\pm0.03}$ | $0.87_{\pm0.03}$ | $0.53_{\pm0.03}$ | $0.39_{\pm0.05}$ | $\mathbf{0.00}_{\pm0.00}$ |
| *PNet (ours)* | $\underline{0.91}_{\pm0.01}$ | $\underline{0.90}_{\pm0.01}$ | $\underline{0.79}_{\pm0.03}$ | $\underline{0.78}_{\pm0.03}$ | $\mathbf{0.00}_{\pm0.00}$ |
| **PNet$^\star$ (ours)** | $\mathbf{0.93}_{\pm0.01}$ | $\mathbf{0.92}_{\pm0.01}$ | $\mathbf{0.87}_{\pm0.05}$ | $\mathbf{0.87}_{\pm0.05}$ | $\mathbf{0.00}_{\pm0.00}$ |
| **Semantic Loss** | | | | | |
| $Baseline_{inD}$ | $0.33_{\pm0.03}$ | $0.22_{\pm0.04}$ | $0.50_{\pm0.00}$ | $0.33_{\pm0.00}$ | $0.57_{\pm0.33}$ |
| $Baseline_{inD,prD}$ | $0.35_{\pm0.05}$ | $0.20_{\pm0.04}$ | $0.50_{\pm0.00}$ | $0.33_{\pm0.00}$ | $0.73_{\pm0.12}$ |
| $C(S)^\star_{inD}$ | $0.39_{\pm0.03}$ | $0.26_{\pm0.06}$ | $0.50_{\pm0.00}$ | $0.34_{\pm0.01}$ | $0.56_{\pm0.25}$ |
| $C(S)^\star_{inD,prD}$ | $0.65_{\pm0.08}$ | $0.60_{\pm0.07}$ | $0.50_{\pm0.00}$ | $0.33_{\pm0.00}$ | $0.36_{\pm0.11}$ |
| $Pre^\star_{inD}$ | $0.58_{\pm0.00}$ | $0.57_{\pm0.00}$ | $0.51_{\pm0.00}$ | $0.43_{\pm0.01}$ | $\mathbf{0.00}_{\pm0.00}$ |
| $Pre^\star_{inD,prD}$ | $\underline{0.91}_{\pm0.01}$ | $0.90_{\pm0.01}$ | $0.77_{\pm0.05}$ | $0.76_{\pm0.06}$ | $\mathbf{0.00}_{\pm0.00}$ |
| $Pre^\star+C(S)^\star_{inD}$ | $0.42_{\pm0.04}$ | $0.36_{\pm0.10}$ | $0.51_{\pm0.03}$ | $0.37_{\pm0.08}$ | $\underline{0.29}_{\pm0.33}$ |
| $Pre^\star+C(S)^\star_{inD,prD}$ | $0.88_{\pm0.02}$ | $0.87_{\pm0.02}$ | $0.50_{\pm0.00}$ | $0.33_{\pm0.00}$ | $\mathbf{0.00}_{\pm0.00}$ |
| **PNet (ours)** | $\mathbf{0.92}_{\pm0.02}$ | $\mathbf{0.92}_{\pm0.02}$ | $\mathbf{0.88}_{\pm0.10}$ | $\mathbf{0.88}_{\pm0.11}$ | $\mathbf{0.00}_{\pm0.00}$ |
| *PNet$^\star$ (ours)* | $\mathbf{0.92}_{\pm0.01}$ | $\underline{0.91}_{\pm0.01}$ | $\underline{0.82}_{\pm0.05}$ | $\underline{0.81}_{\pm0.05}$ | $\mathbf{0.00}_{\pm0.00}$ |
| **DeepProbLog** | | | | | |
| $Baseline_{inD}$ | $0.35_{\pm0.02}$ | $0.26_{\pm0.03}$ | $0.50_{\pm0.01}$ | $0.37_{\pm0.07}$ | $0.52_{\pm0.17}$ |
| $Baseline_{inD,prD}$ | $0.39_{\pm0.05}$ | $0.25_{\pm0.06}$ | $0.50_{\pm0.01}$ | $0.33_{\pm0.00}$ | $0.66_{\pm0.18}$ |
| $C(S)^\star_{inD}$ | $0.37_{\pm0.00}$ | $0.36_{\pm0.00}$ | $0.50_{\pm0.01}$ | $0.34_{\pm0.03}$ | $0.47_{\pm0.08}$ |
| $C(S)^\star_{inD,prD}$ | $0.73_{\pm0.05}$ | $0.68_{\pm0.05}$ | $0.71_{\pm0.21}$ | $0.62_{\pm0.29}$ | $0.37_{\pm0.10}$ |
| $Pre^\star_{inD}$ | $0.58_{\pm0.00}$ | $0.58_{\pm0.00}$ | $0.51_{\pm0.00}$ | $0.44_{\pm0.00}$ | $\mathbf{0.00}_{\pm0.00}$ |
| $Pre^\star_{inD,prD}$ | $\underline{0.92}_{\pm0.00}$ | $\underline{0.91}_{\pm0.00}$ | $\underline{0.81}_{\pm0.01}$ | $\underline{0.80}_{\pm0.01}$ | $\mathbf{0.00}_{\pm0.00}$ |
| $Pre^\star+C(S)^\star_{inD}$ | $0.42_{\pm0.02}$ | $0.41_{\pm0.01}$ | $0.50_{\pm0.01}$ | $0.40_{\pm0.02}$ | $\underline{0.17}_{\pm0.06}$ |
| $Pre^\star+C(S)^\star_{inD,prD}$ | $0.79_{\pm0.08}$ | $0.76_{\pm0.09}$ | $0.61_{\pm0.17}$ | $0.50_{\pm0.25}$ | $0.38_{\pm0.16}$ |
| **PNet (ours)** | $\mathbf{0.94}_{\pm0.00}$ | $\mathbf{0.94}_{\pm0.01}$ | $\mathbf{0.99}_{\pm0.00}$ | $\mathbf{0.99}_{\pm0.00}$ | $\mathbf{0.00}_{\pm0.00}$ |
| **PNet$^\star$ (ours)** | $\mathbf{0.94}_{\pm0.00}$ | $\mathbf{0.94}_{\pm0.01}$ | $\mathbf{0.99}_{\pm0.00}$ | $\mathbf{0.99}_{\pm0.00}$ | $\mathbf{0.00}_{\pm0.00}$ |

Table 11: Mitigation strategies compared for CCN$^+$ on `MNIST-EvenOdd`. Best results in bold and second best underlined.

| | Acc(C)($\uparrow$) | F1(C)($\uparrow$) | Acc(Y)($\uparrow$) | F1(Y)($\uparrow$) | Cls(C)($\downarrow$) |
|---|---|---|---|---|---|
| *Baseline* | $0.03_{\pm0.07}$ | $0.01_{\pm0.03}$ | $0.92_{\pm0.10}$ | $0.92_{\pm0.11}$ | $0.73_{\pm0.05}$ |
| $C(S)^\star$ | $0.36_{\pm0.10}$ | $0.24_{\pm0.08}$ | $0.91_{\pm0.06}$ | $0.90_{\pm0.07}$ | $0.52_{\pm0.11}$ |
| $Pre^\star$ | $0.33_{\pm0.11}$ | $0.22_{\pm0.10}$ | $0.91_{\pm0.09}$ | $0.91_{\pm0.09}$ | $0.57_{\pm0.09}$ |
| $Pre^\star+C(S)^\star$ | $0.54_{\pm0.21}$ | $0.91_{\pm0.10}$ | $0.92_{\pm0.09}$ | $0.91_{\pm0.10}$ | $0.34_{\pm0.22}$ |
| $EPre^\star$ | $0.66_{\pm0.14}$ | $0.56_{\pm0.17}$ | $\mathbf{0.97}_{\pm0.03}$ | $\mathbf{0.97}_{\pm0.03}$ | $\underline{0.25}_{\pm0.16}$ |
| *PNet (ours)* | $\underline{0.96}_{\pm0.04}$ | $\underline{0.95}_{\pm0.05}$ | $\underline{0.96}_{\pm0.07}$ | $\underline{0.96}_{\pm0.08}$ | $\mathbf{0.00}_{\pm0.00}$ |
| *PNet$^\star$(ours)* | $\mathbf{0.98}_{\pm0.01}$ | $\mathbf{0.97}_{\pm0.01}$ | $\underline{0.96}_{\pm0.09}$ | $\underline{0.95}_{\pm0.11}$ | $\mathbf{0.00}_{\pm0.00}$ |

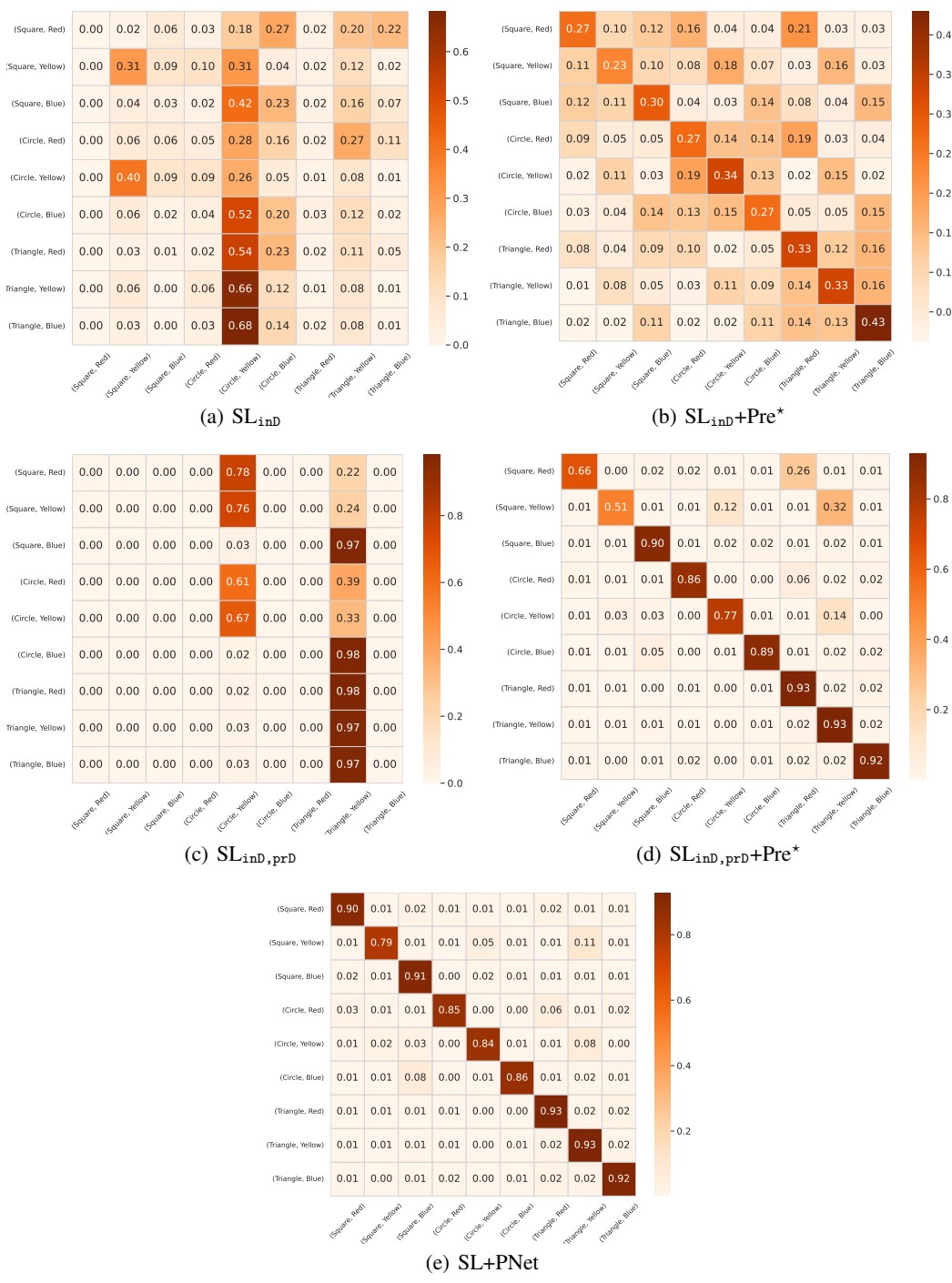

Figure 11: `Kand-Logic` (*shape*,*colour*) concept confusion matrices for the standard and pretrained Semantic Loss model disentangled wrt. inputs (`inD`), inputs and the predicates (`inD,prD`) and integrating prototypes (PNet).

Table 12: Standard mitigation strategies compared for ABL on `MNIST-EvenOdd`.

| | Acc(C)($\uparrow$) | F1(C)($\uparrow$) | Acc(Y)($\uparrow$) | F1(Y)($\uparrow$) | Cls(C)($\downarrow$) |
|---|---|---|---|---|---|
| *Baseline* | $0.06_{\pm 0.09}$ | $0.03_{\pm 0.05}$ | $0.70_{\pm 0.08}$ | $0.31_{\pm 0.14}$ | $0.56_{\pm 0.15}$ |
| *C(S)$^\star$* | $0.67_{\pm 0.16}$ | $0.64_{\pm 0.16}$ | $0.71_{\pm 0.10}$ | $0.15_{\pm 0.02}$ | $0.00_{\pm 0.00}$ |
| *Pre$^\star$* | $0.90_{\pm 0.03}$ | $0.89_{\pm 0.02}$ | $0.83_{\pm 0.04}$ | $0.15_{\pm 0.01}$ | $0.00_{\pm 0.00}$ |
| *Pre$^\star$+C(S)$^\star$* | $0.95_{\pm 0.01}$ | $0.93_{\pm 0.01}$ | $0.90_{\pm 0.01}$ | $0.17_{\pm 0.01}$ | $0.00_{\pm 0.00}$ |

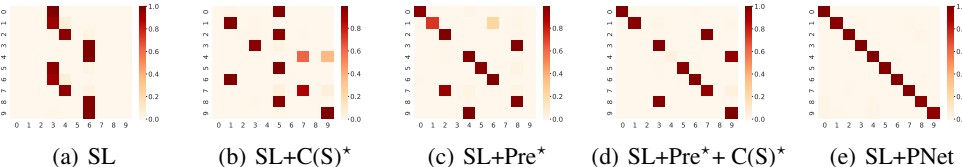

(a) SL     (b) SL+C(S)$^\star$     (c) SL+Pre$^\star$     (d) SL+Pre$^\star$+ C(S)$^\star$     (e) SL+PNet

Figure 12: `MNIST-EvenOdd` concept confusion matrices for different mitigation strategies.

Crucially, even though concept-level scores for this mitigation are comparable to PNet models, the performance on label-level metrics is lower than the one of our approach. This can be explained by the fact that the embedding update introduced in Section 4 does a better job than the standard NeSy update in aligning the concept representation with the final label prediction task.

The improved concept alignment Nesy models enjoy when paired with prototypical extractors is even more clear looking at the confusion matrices: in Figure 12 we see how different mitigation strategies allow to gradually improve the Semantic Loss model ability to correctly reconstruct the unlabelled concepts, with the prototypical model being the better. Similar considerations hold for the confusion matrices reported in Figure 11 for the case of `Kand-Logic`: we can see that incorporating the disentanglement wrt. the shape and colour concepts does not help the Semantic Loss model to better disambiguate the concepts, but the difference between the two is evident as soon as the support set is made available for pretraining. Even in the case of a pretrained, fully disentangled model however, there concept $\square$ is often confused with $\triangle$. On the other hand, the prototypical NeSy model does not suffer for this confusion.

Lastly, we report the full results for `BDD-OIA` in Table 9. In this task the data are very unbalanced (c.f. Table 7). Interestingly, the best results our architecture achieves for F1(C) and Cls(C) show that the prototypical model is robust wrt. the unbalance and does not collapse its predictions on the dominant class "0". Conversely, pretraining on the support set not help in mitigating the shortcut reasoning behaviour. Even worse, the score for F1(C) decreases for pretrained models wrt. the standard baselines. Overall, the most competitive approach with prototypical models is C(S)$^\star$ in its twofold disentangled version. However, this baseline as well only achieves a minor improvement wrt. the F1(C) score (0.11) of the standard DeepProbLog model (0.08). Conversely, the PNet model doubles this score (0.16). In addition, even if the C(S)$^\star$ baseline succeeds in reducing concept collapse (0.51), it is still far from the lower collapse of the prototypical model (0.35). We observe that label-level metrics and Acc(C) decrease in the PNet models: this is expected as, given the unbalanced data, baselines frequently predicting class "0".

**RQ2.** *How does prototypical NeSy compare with other mitigation strategies?*

In Table 8 we report an exhaustive picture for the comparison of shortcut mitigation strategies in `MNIST-EvenOdd`. The Table accounts for the mitigations introduced in [29] already discussed in the main text (Section 6 (RQ2)): these are *partial concept supervision* (C), *input reconstruction* (R) via the introduction of a reconstruction loss with an encoder-decoder architecture, *Shannon entropy* loss (H), and combinations thereof. The scores for baseline models are obtained from [5] and we used the code for all the mitigation strategies and combinations thereof as in [29]. Also, to convey the broader picture, we included the computation for concept (Acc(C)) and label (Acc(Y)) accuracy. As expected, we see that the results wrt. Acc(C) mirror those reported in Table 4, where we reported the F1(C) for the same experiments. Moreover, our models consistently succeed in minimizing Cls(C), which is 0.0 across all the models considered. On the other hand, we see that the mitigation strategies generally

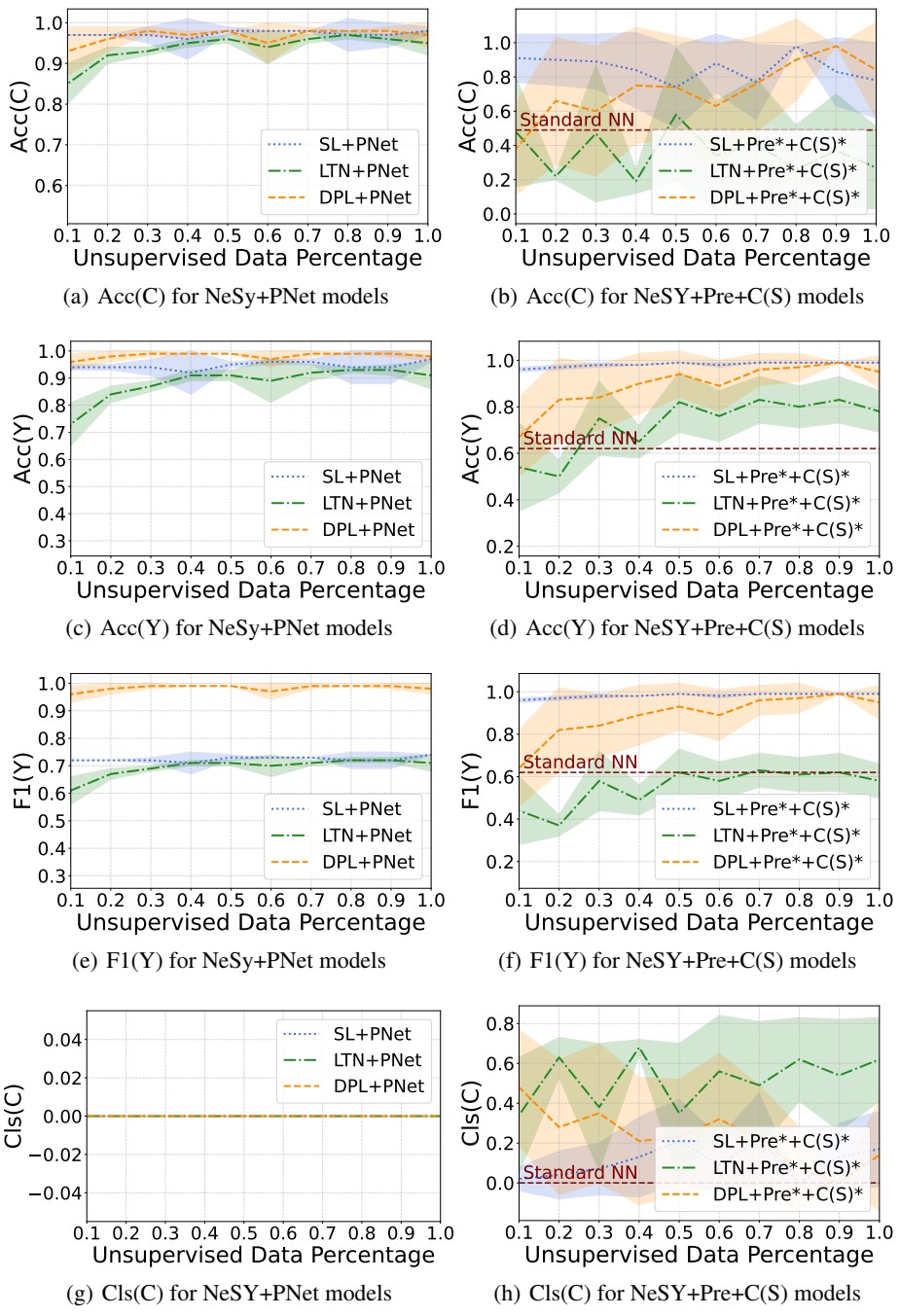

(a) Acc(C) for NeSy+PNet models

(b) Acc(C) for NeSY+Pre+C(S) models

(c) Acc(Y) for NeSy+PNet models

(d) Acc(Y) for NeSY+Pre+C(S) models

(e) F1(Y) for NeSy+PNet models

(f) F1(Y) for NeSY+Pre+C(S) models

(g) Cls(C) for NeSY+PNet models

(h) Cls(C) for NeSY+Pre+C(S) models

Figure 13: Extensive Comparison of Prototypical NeSy models (left) vs Standard Pretrained ones (right) supervised on support sets, across different percentages of unlabelled datapoints on `MNIST-EvenOdd`.

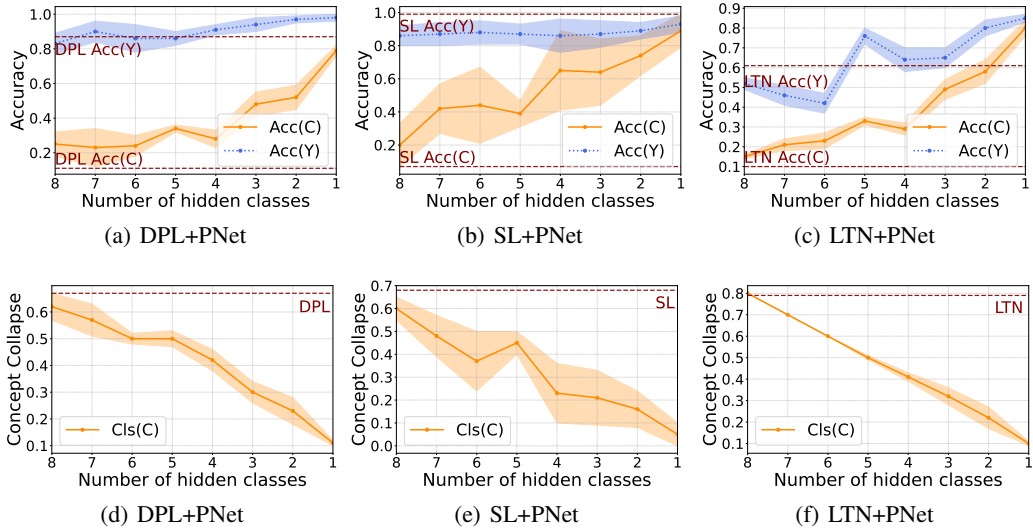

Figure 14: Acc(C), Acc(Y) and Cls(C) obtained with the Prototypical NeSy models on `MNIST-EvenOdd` when varying the number of hidden classes, i.e., the number of concepts for which we have zero examples. The red dotted lines indicate the performance of the corresponding baseline model.

do not have a great impact on the metrics over the final label (i.e., Acc(Y) and F1(Y)), as the NeSy models were already performing well on the unlabelled classes.

**RQ3.** *Are Prototypical NeSy models robust wrt. the number of unlabelled datapoints?*

In Figure 13 we report the results obtained in the same experiment reported in Section 6 (RQ3) wrt. the metrics Acc(C), Acc(Y), F1(C) and F1(Y). Similarly to F1(C), we can see from the Figure (a) that the results we obtain for ACC(C) using our PNets are very stable and improve as we add more unsupervised data. On the contrary, as (b) shows, if we use the standard NeSy models taking advantage of the supervisions both in a pretraining stage (*Pre*) and at training time with a cross-entropy loss (*C(S)*), the results present a high variance and are very susceptible to the percentage of unsupervised data used for training. Similar considerations hold for ACC(Y) as well in (c) and (d), with only the Semantic Loss baseline model maintaining label accuracy scores comparable with the PNet approach. Moreover, from (e) and (f) we notice the PNet approach consistently improves the standard NeSy models F1(Y) score as well (with SL being the only exception) for all the percentages of unsupervised data. Crucially, observe from (h) that all the standard models suffer from high concept collapse. How much each model collapses it predictions is again susceptible to the percentage of unsupervised data used for training. Now compare this behaviour with (g): our prototypical models have 0.0 concept collapse across *all* the percentages of unsupervised data.

**RQ4.** *What is the impact of having unlabelled classes?*

In Figure 14 and Figure 15 we report the results obtained in the experiments described in Section 6 (RQ4) wrt. Acc(C), Acc(Y) and Cls(C) on the `MNIST-EvenOdd` task and the `Kand-Logic` task, respectively. As expected, the more classes are labelled the better the results become. However, it is interesting to note that—in the `MNIST-EvenOdd` task—even by labelling only two images we obtain better results over the concepts than using the standard NeSy models. The same happens for the `Kand-Logic` task, where we need at least two examples for the shape primitives and two for the colour primitives. However, we can again see that with just four labelled examples we get better results than all the standard NeSy models.

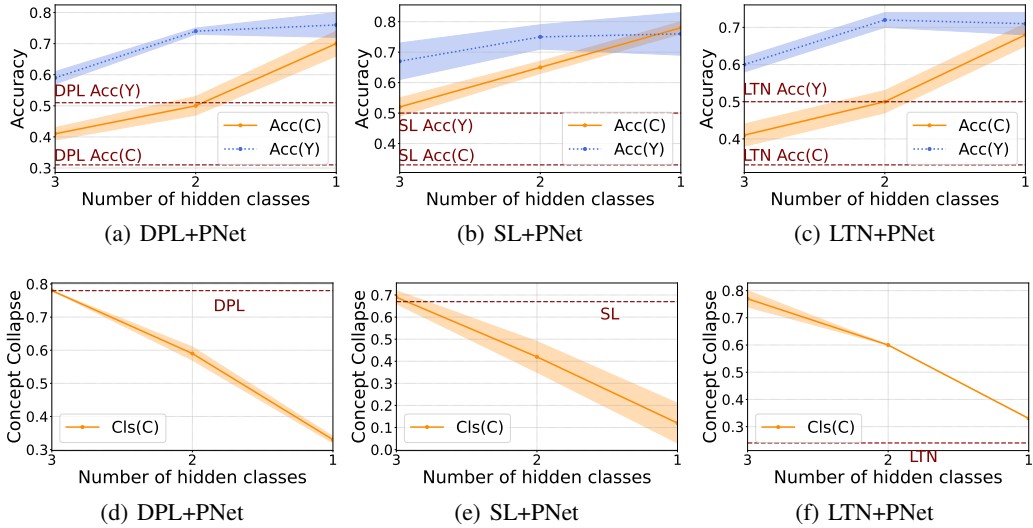

Figure 15: Acc(C), Acc(Y) and Cls(C) obtained with the Prototypical NeSy models on `Kand-Logic` when varying the number of hidden classes, i.e., the number of concepts for which we have zero examples. The red dotted lines indicate the performance of the corresponding baseline model.

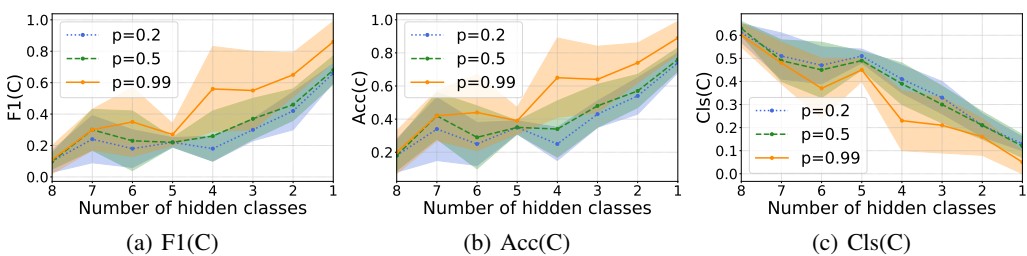

Figure 16: F1(C), Acc(C) and Cls(C) obtained with PNet+SL when varying the number of hidden classes and for different values of $p$, i.e., for different lower-tail quantiles of the $\chi^2$ distribution.

**RQ5:** *How important is the centroid initialization for unlabelled classes?*

In Figure 16 we report the results obtained wrt. F1(C), Acc(C) and Cls(C) for different lower-tail quantiles of the $\chi^2$ distribution. Clearly, choosing $p = 0.99$ improved our results—no matter the metric considered.

