# OpenReview forum: "Right for the Right Reasons: Avoiding Reasoning Shortcuts via Prototypical Neurosymbolic AI"
_NeurIPS.cc/2025/Conference — NeurIPS 2025 poster_

### Official Review · Reviewer_CSxp · 2025-06-08

**Clarity:** 3
**Significance:** 2
**Originality:** 2
**Rating:** 4
**Confidence:** 4

**Summary:**

This paper proposes using prototype-augmented neural networks to mitigate reasoning shortcuts. In the proposed method, prototype extractors with learnable parameters first compute class centroids from few labeled examples. A neural network is then trained to classify datapoints based on their proximity to the corresponding centroids while simultaneously satisfying background knowledge constraints.

The proposed method can be used in conjunction with other NeSy models, such as DPL, LTN, and SL. Experiments show that the proposed method can significantly enhance these NeSy approaches, achieving F1-scores of 98% on MNIST-EvenOdd and 94% on Kand-Logic, even when only one labeled datapoint is provided for each concept.

**Questions:**

1. Consider a direct, intuitive strategy: first, use an SSL or one-shot learning method to predict labels for the unlabeled data, and then provide these pseudo-labels to a NeSy algorithm. What are the differences and potential advantages of the proposed prototype-augmented strategy compared to this approach?

I would be willing to raise my score if the authors could provide convincing evidence that their algorithm is substantially different from, and outperforms, more straightforward counterparts.

Such a comparison would be necessary to fully support the claimed novelty of their work.

**Ethical Concerns:**

["NO or VERY MINOR ethics concerns only"]

**Final Justification:**

The authors provide additional experimental results and introduce new baseline models for comparison during rebuttal. These updates have addressed most of my concerns. However, some areas could still be clarified, such as the specific conditions or assumptions under which PNet is expected to succeed or fail.

Given the substantial improvements, I am raising my score.

**Limitations:**

As mentioned in the weaknesses section, the proposed approach assumes that concepts form well-separated clusters. This property may not hold for more realistic NeSy tasks.

**Quality:**

3

**Strengths And Weaknesses:**

**Strengths**

1. To my knowledge, the idea of introducing prototypical neural networks into NeSy research is novel and interesting.
2. The problem setting is solid and reasonable. The assumption of having one or a few labeled examples per class is intuitive and practically feasible.
3. The experimental results are quite impressive, as reasoning shortcuts are almost completely avoided on the tested benchmarks, especially MNIST-EvenOdd and Kand-Logic.
4. The proposed method can be easily incorporated into other NeSy models, thus extending its practical usefulness.

**Weaknesses**

1. The methodology of this work seems to heavily overlap with existing NeSy research that utilizes Semi-Supervised Learning (SSL) techniques, but this relationship is not discussed in detail in the manuscript.
    - The proposed method is primarily based on the assumption that "datapoints close to each other are more likely to belong to the same concept," which is known as the cluster assumption in SSL research.
    - To my knowledge, some existing NeSy research already utilizes SSL techniques, either explicitly or implicitly (e.g., by using distance-based metrics between datapoints in loss functions). However, most experimental results reported in the manuscript (except Table 2) are based only on the proposed method or its ablations; a proper comparison with more relevant methods would be helpful.
    - From an SSL perspective, the proposed framework is quite similar to a process of: 1) using self-training to assign labels to unlabeled datapoints by first initializing centroids from labeled data, assigning pseudo-labels based on distance, and iterating until convergence; and then 2) using this predicted label information to enhance NeSy models. Consequently, the novelty of the proposed method appears somewhat limited.

2. The crucial cluster assumption may not always hold in typical NeSy application scenarios, which relates to the authors' discussion in Section 7 regarding the method's sensitivity to ambiguities in concept definition.
    - IMHO, a major motivation for using NeSy methods is for scenarios where data alone is insufficient—for example, with very noisy data. In such cases, one hopes a NeSy system can perform better by utilizing symbolic knowledge.
    - This leads to a paradox: if data is clean and can be easily classified, neural networks are sufficient, and there's no need for NeSy; if data is noisy or concept definitions are ambiguous, NeSy is needed, but the proposed method would not work well in this case. This raises the question of when we should expect to use it.

---

> ### Author Rebuttal · Authors · 2025-07-30
>
> Firstly, we would like to thank the reviewer for their comments, which allowed us to highlight better both the novelty of our approach and its broad applicability. Additionally, we are thankful for pointing us towards the abductive learning literature, if we have missed some reference, please let us know.
>
> > [W1.a] Same assumption as SSL+Nesy (abductive learning) methods
>
> Our method relies on a slightly different assumption: namely, that datapoints are similar—i.e., clusterable—in the learned embedding space. The embedding space is not fixed but is shaped dynamically during training via backpropagation from both the supervision provided by labelled datapoints and the neuro-symbolic (NeSy) loss. Consequently, similarity in this space does not necessarily correspond to similarity in the original input (feature) space. Rather, it reflects a form of functional similarity that integrates both the data and the symbolic knowledge expressed in the constraints. In Section 4, we formally derive the gradient of the total loss, showing that the embeddings are updated based on both (i) their proximity to class prototypes and (ii) the error signal stemming from the NeSy component. This confirms that the learned representation is shaped by the interplay between data-driven and knowledge-driven signals.
>
> Whenever pseudo-labels are involved, it is indeed a core problem to decide how and when to commit to a label, as an early pseudo-labelling mistake might propagate further. This is why in the abductive learning community, many researchers have studied how to define consistency measures to update the candidate pseudolabels with the most similar ones which are also compliant with the background knowledge. For example, [1,2] take the consistency score as the size of the largest subset of unlabeled examples with abduced labels that are consistent with the knowledge base, while [4,5,6] measure
> the confidence of the predicted labels by the machine learning model.
>
> The most similar work to ours in spirit is ABLSim [7], as they also use the embedding similarity to guide training. However, ABLSim constructs a similarity-based consistency score purely in the embedding space, encouraging samples with identical (pseudo-)labels to form tight clusters and those with differing labels to be well-separated. This score is used to select among logically consistent pseudo-label assignments, and the perception model is trained accordingly. In contrast, our method integrates symbolic knowledge directly into the loss function, such that the embedding space is shaped not only by class-based proximity (via prototype loss) but also by gradients arising from differentiable neuro-symbolic constraints. This results in an embedding space that reflects functional similarity: two points may be embedded nearby not because they share superficial input features, but because they induce similar outcomes with respect to both the downstream task and the symbolic knowledge. As shown in our gradient derivations (Section 4), this joint signal ensures that the learned representation is aligned with the structure of both the data and the knowledge base—beyond what similarity in feature space alone can capture. We will integrate this discussion in the paper.
>
>
> > [Q1+W1.b] Comparison with SSL+NeSy methods or intuitive strategy
>
> We thank the reviewer for their suggestion, and to answer to their question we compare our method against one abductive learning method [1-7], that we believe is a much stronger baseline than the intuitive strategy.
>
> In particular, we trained this model using the ABLKit package [3] on MNIST-EvenOdd, (i) first without giving the model any labelled datapoint (indicated with ABL in Table A below) and then (ii) giving the model access to exactly the same augmented labelled datasets as our models (indicated as ABL (Aug) in Table A below).  In Table A we report the results. For comparison, we also report the performance of our DPL+PNet.
>
> Table A. Average results obtained on MNIST-EvenOdd over 5 seeds.
> |Model|Acc(Y)|Acc(C)|
> |--|--|--|
> |ABL|0.70±0.05|0.12±0.12|
> |ABL(Aug)|0.72±0.07|0.51±0.24|
> |DPL+PNet|**0.99±0.00**|**0.98±0.00**|
>
> As we can see from the results, the semi-supervised method is able to improve its own performance by using the augmented datasets, but is not able to perform as well as the prototypical networks. The much higher standard deviation presented by the abductive learning method is also something worth noticing: when performing pseudo-labelling, the methods have to “commit” to a choice (e.g., by classifying an image of the digit 7 as indeed 7 or as 9):, if the wrong decision is taken early on, then this error will propagate through training.
>
> More in general, notice that ABLKit obtains very good performance on the standard MNIST-Add dataset, where it achieves Acc(C) = 0.98, thus outperforming both LTN and DeepProbLog. However, MNIST-EvenOdd is a particularly challenging scenario due to the high number of reasoning shortcuts it presents. Indeed, even assuming the model achieves perfect final label accuracy (Acc(Y) = 1.00), there remain 49 possible digit-level assignments it could have produced, of which only one corresponds to the correct underlying reasoning. This highlights the difficulty of the task: neither the knowledge nor the data alone are enough to learn the right concepts.
>
> > [W1.c] The method proposed is very similar to training with SSL or one-shot learning until convergence and then using this predicted label information to enhance NeSy models.
>
> Our approach is similar in spirit to the method proposed by the reviewer but actually it very presents some fundamental differences highlighted by the following example. Suppose we have (as in our case) a single example for each concept (even augmented) and training a SSL or one-shot learning method until convergence. The probability of the network overfitting on the few available examples will be very high. If then we use the overfitted network to train with the NeSy methods, then each network update will only take into consideration the error with respect to the NeSy loss thus making it very likely that the model will again fall prey to the reasoning shortcuts problem. On the contrary, the weights update done in our method takes into account both embedding similarity and knowledge satisfaction for all datapoints (both labelled and unlabelled) -- see Theorem 4.1 and Corollary 4.2.
>
> > [W2] NeSy methods should perform well in noisy scenarios
>
> We totally agree with the reviewer. Given your response and those of other reviewers, we realised though that we might have overstated the limitations of the approach. To obviate with such problem, we run the following models on the autonomous driving task (BDD-OIA) and report the results obtained in Table B:
>
> 1. A fully supervised model, which has been trained on all images and all their lab (Fully Supervised in the Table),
> 2. An augmented version of DeepProbLog, which has been trained with all the unsupervised images and the 128 labelled datapoints we used to train our model (DPL (Aug) in the Table),
> 3. A new version of our model, where we use prototypes for all concepts, and not just those that cause the action “Stop” (DPL+PNet in the Table).
>
> For completeness, we also include the results with the standard DPL method, which has been trained with no labelled images.
>
> Table B. Results on the autonomous driving dataset (BDD-OIA)
>
> |Model|F1(C) ($\uparrow$)|Cls(C) ($\downarrow$)|
> |--|--|--|
> |DPL|0.04±0.00|0.85±0.01|
> |DPL (Aug)|0.07±0.07|0.75±0.05|
> |Fully Supervised|0.06±0.06|0.78±0.10|
> |DPL+PNet|**0.16±0.15**|**0.35±0.01**|
>
> As we can see from the results, even though the improvements are smaller than in the MNIST-EvenOdd and Kand-Logic, using DeepProbLog and our PNets we manage to outperform even the fully supervised model, trained with all the supervised images available in the dataset. To train DPL+PNet on the other hand, we only had to label only 128 images, with no data augmentation performed.
>
> If we compare the results concept by concept for the fully supervised model and DPL+PNets, we can see that for almost all the concepts DPL+PNet outperforms the fully supervised model.
>
> Table C. Concept-level performance (in terms of F1) on BDD-OIA
>
> |Concept|Fully Supervised|DPL+PNet|
> |--|--|--|
> |Green Light|16.66|**46.01**|
> |Follow Traffic|25.39|**30.34**|
> |Road is clear|3.70|**28.53**|
> |Red Light|**9.41**|5.36|
> |Traffic Sign|9.51|**21.31**|
> |Obstacle Car|0.55|**3.21**|
> |Obstacle Person|0.00|**1.16**|
> |Obstacle Rider|6.48|**23.27**|
> |Obstacle Other|0.29|**2.37**|
> |No Lane Left|0.00|**0.31**|
> |Obstacle on Left|0.00|**3.56**|
> |Solid Line Left|0.00|**1.41**|
> |On Right Turn Lane|0.10|**1.11**|
> |Traffic Light Allows Right|0.74|**7.18**|
> |Front Car Turning Right|0.30|**3.18**|
> |No Lane Right|4.40|**31.72**|
> |Obstacle on Right|3.29|**32.17**|
> |Solid Line Right|3.84|**26.69**|
> |On Left Turn Lane|27.98|**38.57**|
> |Traffic Light Allows Left|5.02|**22.92**|
> |Front Car Turning Left|5.57|**9.59**|
>
> This is indeed in line with the assumption of the reviewer, that in presence of noisy data a NeSy system with access to very limited data can perform better than a fully data-driven one.
>
> ### References:
>
> [1] Wang-Zhou Dai et al., Bridging Machine Learning and Logical Reasoning by Abductive Learning, Neurips, 2019
>
> [2] Zhi-Hua ZHOU, Abductive learning: towards bridging machine learning and logical reasoning, 2019
>
> [3] Yu-Xuan Huang et al., ABLkit: a Python toolkit for abductive learning, Frontiers of Computer Science, 2024
>
> [4] Qing Li et al., Closed loop neural-symbolic learning via integrating neural perception, grammar parsing, and symbolic reasoning. ICML, 2020.
>
> [5] Wang-Zhou Dai and Stephen H. Muggleton. Abductive knowledge induction from raw data. IJCAI, 2021.
>
> [6] Le-Wen Cai et al. Abductive learning with ground knowledge base. IJCAI, 2021
>
> [7] Yu-Xuan Huang et al., Fast Abductive Learning by Similarity-based Consistency Optimization, NeurIPS, 2024

---

> > ### Comment · Reviewer_CSxp · 2025-08-06
> >
> > I thank the authors for providing additional experimental results and introducing new baseline models for comparison. These updates have addressed most of my concerns. However, some areas could still be clarified, such as the specific conditions or assumptions under which PNet is expected to succeed or fail.
> >
> > Given the substantial improvements, I am raising my score.

---

> ### Comment · Area_Chair_w3ks · 2025-08-05
> **Ping**
>
> Dear Reviewer,
>
> The deadline for the author-reviewer discussion is approaching (Aug 8, 11.59pm AoE).
> Please read carefully the authors' rebuttal and engage in meaningful discussion.
>
> Thank you,
> Your AC

---

### Official Review · Reviewer_cxP2 · 2025-06-23

**Clarity:** 3
**Significance:** 2
**Originality:** 2
**Rating:** 4
**Confidence:** 3

**Summary:**

The paper proposes combining neuro-symbolic predictors with prototype-augmented neural networks to mitigate reasoning shortcuts. The central idea is to begin training with a minimal set of annotated prototypes. The approach is validated through experiments that demonstrate its effectiveness and analyze concept collapse as the number of supervised prototypes varies.

**Questions:**

* The method appears to rely heavily on initialization, particularly on having predefined prototypes. In standard disentanglement settings, wouldn’t this require one prototype per concept configuration? For example, in MNIST-Even-Odd with 10 digits per position, this would imply annotating 100 configurations. Have you explored scenarios where disentanglement does not hold in terms of required annotations?

* If prototypes are learned from data, there’s a risk they may collapse to similar representations—i.e., the model may predict different concepts, but their prototypes end up qualitatively indistinguishable. Have you observed such behavior during training?

* A key assumption seems to be that concepts are clusterable in the representation space. Could you elaborate on this requirement and its implications for generalization and applicability?

* Relatedly, have you considered a theoretical analysis of how the use of prototypes impacts the number of reasoning shortcuts that can affect the model?

* How does the method perform in fully unsupervised settings where no prototypes are available, i.e., the standard NeSy setup? You report some interesting results in the appendix (i.e., hidden class count and the BDD-OIA analysis), which seem impactful enough to highlight in the main text.

* You mention that the solution proposed in [1] requires dense annotations across many datapoints, which may be costly. Could you clarify why this is the case? It would also be interesting to compare its annotation efficiency with that of prototype-based networks as well as for other mitigation strategies.

* The code is currently unavailable. Could you clarify the reason for this?

-----

[1] E. Marconato, BEARS Make Neuro-Symbolic Models Aware of their Reasoning Shortcuts, in UAI (2024)

**Ethical Concerns:**

["NO or VERY MINOR ethics concerns only"]

**Final Justification:**

Rebuttal

**Limitations:**

Yes

**Quality:**

2

**Strengths And Weaknesses:**

**Strengths**: The paper addresses an important problem in the literature with a clear and well-motivated idea. It includes a wide range of experiments and research questions, and it explicitly discusses the limitations of the proposed approach.

**Weaknesses**: Effectively mitigating RSs with a prototype-based network requires a well-initialized prototype for each concept configuration, and assumes that concepts form clear clusters. The zero-annotation setting is explored with less focus (there are interesting experiments in the appendix), and there is no theoretical analysis of how prototypes influence the number of reasoning shortcuts.

---

> ### Author Rebuttal · Authors · 2025-07-30
>
> We thank the Reviewer for all the comments, and especially for encouraging us to consider a theoretical analysis of how the use of prototypes impacts the number of reasoning shortcuts that can affect the model. We believe this will be a great contribution to the paper.
>
> > [Q1] Reliance on prototype initialization and consideration of scenarios where disentanglement does not hold in terms of required annotations
>
> We believe there might have been a slight misunderstanding here. We will make sure the below will be made clear in the paper. Regarding the reliance on prototype initialization, our method relies on prototype initialization as much as standard networks rely on weights initialization. In the paper we have simply provided a smart initialization scheme which allows us to avoid the trivial satisfaction of the constraints. This initialization scheme can be applied in any scenario.
>
> Additionally, please note that our method does not require an example for every possible configuration, but rather one example per concept. For example, in the case of MNIST-EvenOdd, for RQ1 we annotated 10 images—one for every digit).
>
> If we have one labelled example for every concept, the prototypes get trained and our method is very stable wrt weights initialization (see, e.g., Table 1), and as expected, if we have 0 labelled examples for more than one concept then we can see  in Figure 5  that the method becomes more noisy. (This chart also shows why it is important to pick $p=0.99$ in our initialization).
>
> When compared to training standard networks with NeSy methods (see Figure 3) we can see that our approach is much more robust to the number of unsupervised datapoints during training, even though the NeSy methods have been trained with exactly the same amount of labelled datapoints.
>
> Regarding the required annotations, in RQ4 we have studied what happens when we do not provide any labelled examples for a subset of the concepts. For example, in Figure 4 we show how the performance vary on the ground of how many classes we leave unspecified in the MNIST-EvenOdd dataset (i.e., how many classes we have in our dataset for which we have 0 labelled examples.) Notice that when we say that we have 8 specified classes, it means that we have only labelled two images and then we perform data augmentation on those images.
>
>
> > [Q2] Representation collapse
>
> We have never experienced centroid collapse. We will provide the t-SNE plots in the paper in case of acceptance.
> In BDD-OIA though, we have seen the representations of all the datapoints clustering around the negative prototypes  (and this is why we have concept collapse $\not = 0$).
>
> > [Q3] Dataset needs to be clusterable in the representation space
>
> We thank the reviewer for this comment, which gave us some ideas for future development of this work. While it is true that prototypical networks assume that each class forms a cluster in the representation space, we emphasize that the model is explicitly trained to learn embeddings in which such clusterability emerges, and in our practical tasks this assumption is empirically well-supported. In general, this assumption is particularly suited for standard NeSy methods, which learn “basic concepts” and then use the logical constraints to infer the more complex ones.
>
> However, we agree that for example in presence of highly multi-modal distributions one cluster might not be enough. To obviate this problem, we could study how to learn multiple prototypes for each class (as done in [1])  and then use the logical constraints.
>
>
>
> > [Q4] Theoretical analysis on the number of reasoning shortcuts
>
> We totally agree that adding this an analysis on the number of reasoning shortcuts would be beneficial. Below we conduct such analysis and we will include it in the paper in case of acceptance.
>
> For clarity, we follow the same notation as [2].
>
> Suppose the conditional distribution $p_\theta(\mathbf{Y} \mid \mathbf{X}; K)$ is induced by a prototypical network with embedding function $f_\theta: \mathbb{R}^d \to \mathbb{R}^{m_1 \times m_2 \times \ldots \times m_k}$.
>
> Assume that the datapoints are clusterable in the embedding space. Now, consider the subset of concepts indexed by $I \subseteq [k]$ for which we have at least one labeled datapoint available. Let $\mathcal{S} \subseteq \text{supp}(G)$ be the subset of concept assignments in the support of $G$ where concept supervision is available (at least one labeled example per concept $i \in I$).
>
> By adding a cross-entropy loss for the concepts in $I$ to the training objective, given the clusterability assumption, and under the Assumption 1 from [2] the number of deterministic $p_\theta(\mathbf{C} \mid \mathbf{G})$ optima becomes:
>
> $$
> \sum_{\alpha \in \mathcal{A}} \mathbb{1}\{
> \bigwedge_{g \in \mathcal{S}} \bigwedge_{i \in I} \alpha_i(g) = g_i
> \}.
> $$
>
> Notice that this is *the same number of deterministic optima obtained by the data-based mitigation strategy in [2], which assumes that for every concept* $i \in I$ and for **every datapoint** *the concept supervision is provided*. Our method achieves this equivalence by leveraging clusterability in embedding space, hence reducing the labeling effort required. Indeed, under our clusterability assumption, it suffices to have only a single labeled datapoint for each concept $i \in I$ to match the theoretical guarantees obtained by dense supervision in [2].
>
> > [Q5] Tested on totally unsupervised settings
>
>  One of the assumptions of this work is that we assume we have at least two datapoints belonging to two different classes (or concepts) labelled. Indeed, as shown above, without such supervision the number of reasoning shortcuts is equivalent to the number of reasoning shortcuts existing in any other model. Also, looking at the results shown in RQ4 and Figure 4 (as well as Figures 13 and 14 in the Appendix), it is clear that the more concepts have at least one labelled example, the better the results. This is a very lightweight assumption in many application domains, as randomly sampling two datapoints and labelling them is not costly. However, we agree that how to drop this assumption is an interesting avenue for the future.
>
> > [Q6] Solution proposed in [3] requires dense annotations
>
> We believe there might have been a slight misunderstanding here.
>
> We stated that one of the mitigation strategies proposed in [2] requires dense annotations. In [2], to conclude Proposition 5, the authors need to have dense annotations for the concepts they want to consider annotated, otherwise they cannot make use the fact that the deterministic concept distributions $p_\theta(\mathbf{C} |\mathbf{X})$ of NeSy predictors $p_\theta(\mathbf{Y} \mid \mathbf{X}; K)$, correspond one-to-one to the deterministic distributions $p_\theta(\mathbf{C} \mid \mathbf{G})$ yielding label distributions $p_\theta(\mathbf{Y} \mid \mathbf{G}; K)$.
>
> In our experimental analysis (RQ2) we have compared with the mitigation strategies proposed so far in the literature and also with their combination. As we can see, using prototypical networks managed to outperform even the combination of the three mitigation strategies proposed so far.
>
> Regarding a study on the efficacy of these methods, we will try to conduct one in the rebuttal time.
>
> > [Q7] Code unavailable
>
> Our paper is not yet publicly available on arxiv. We will make the code and paper publicly available upon publication.
>
> ### References:
>
> [1] Allen et al. Infinite Mixture Prototypes for Few-Shot Learning, 2019.
>
> [2] Marconato et al. Not All Neuro-Symbolic Concepts Are Created Equal: Analysis and Mitigation of Reasoning Shortcuts, 2023
>
> [3] E. Marconato, BEARS Make Neuro-Symbolic Models Aware of their Reasoning Shortcuts,  UAI (2024)

---

> ### Comment · Reviewer_cxP2 · 2025-08-04
>
> Thank you for the detailed response.
>
> I have a few additional concerns. I may not have expressed myself clearly earlier, please feel free to ask for clarification if that is the case.
>
> First, while I agree that only one labeled concept is needed under the **disentanglement** assumption, this assumption is critical. If it does not hold, you would probably need a labeled example for each possible concept configuration. For datasets like BDD-OIA, this could require up to $2^{21}$ labelled concept configurations, which seems impractical. Could elaborate on this?
>
> Secondly, regarding your answers to Q1, Q3, and Q6: I believe the main text would benefit from clearer explanations on those points.
>
> Lastly, regarding the answer for Q5, I found the BDD-OIA experiment in the appendix particularly compelling. This is just general curiosity, do you think it could fit into the main paper?

---

> > ### Author Response · Authors · 2025-08-04
> >
> > Dear Reviewer,
> >
> > First of all, we are deeply thankful for engaging with us and for clarifying your question. We are also happy you appreciated our answers so far.
> >
> > Regarding the disentanglement assumption, we will make it explicit in the paper that our architecture is disentangled by design, thereby emphasizing its central role. As shown in [1], this reduces the number of reasoning shortcuts, which is precisely why we require only one labeled example per concept—not per full concept configuration—to train our prototypical network effectively.
> >
> > Indeed, following [1]'s notation, we partition the available set of concepts in $k$ subsets of mutually exclusive concepts, and then we create an independent prototype extractor for each one of them.  Each prototype extractor induces a distribution $p^j_{\theta}(C_j \mid G_j)$, and the overall probability can be calculated as $\prod_{j \in [k]} p^j_{\theta}(C_j \mid G_j)$, this corresponds exactly to the disentanglement definition given in Section 5.4 of [1].
> >
> > Regarding answers to Q1, Q3 and Q6: we will incorporate the above explanations on those points.
> >
> > Regarding the BDD-OIA experiments, we agree they strengthen our results and will move them to the main body of the paper. To make room, we will likely merge the discussions and tables for RQ1 and RQ2. This is especially important given the updated (and improved) results on BDD-OIA (see our response to reviewer CSxp, W2).
> >
> > If these clarifications address your concerns, we would be very grateful if you would consider updating your score.
> >
> > ### References
> >
> > [1] Marconato et al. Not All Neuro-Symbolic Concepts Are Created Equal: Analysis and Mitigation of Reasoning Shortcuts, 2023

---

> > > ### Comment · Reviewer_cxP2 · 2025-08-05
> > >
> > > Thank you for addressing my final concerns. I am therefore updating my score to reflect the improvements.

---

### Official Review · Reviewer_wXk3 · 2025-07-02

**Clarity:** 3
**Significance:** 2
**Originality:** 2
**Rating:** 3
**Confidence:** 4

**Summary:**

The paper addresses the problem of "reasoning shortcuts" in Neurosymbolic (NeSy) models, where a model might satisfy symbolic constraints syntactically but fail to learn the correct underlying semantics of concepts. The authors propose a Prototype-Augmented NeSy model, which integrates prototypical networks into standard NeSy frameworks. The core idea is to anchor the latent representations of concepts to prototypes learned from a very small number of labeled examples. The authors conduct experiments on the rs-bench benchmark, demonstrating improvements in concept learning accuracy and a significant reduction in reasoning shortcuts.

**Questions:**

1.How does prototype proximity translate into logical consistency? Can the authors formally explain why minimizing the distance between an embedding and a prototype should lead to better satisfaction of logical constraints?
2.What is the precise relationship between semantic alignment and symbolic satisfaction? The authors use the term “semantic alignment” extensively, but it is never formally defined. How is it measured, and how does it relate to the logical correctness of concept assignments?

**Ethical Concerns:**

["NO or VERY MINOR ethics concerns only"]

**Final Justification:**

I thank the authors for their response. While some concerns have been addressed, this work still requires significant improvement to meet NeurIPS standards. It is not yet ready for publication, as substantial additional work is needed. I will maintain my original score.

**Limitations:**

The limitations of this work primarily concern the scope and depth of the evaluation; see details in the weaknesses section.

**Quality:**

2

**Strengths And Weaknesses:**

Strength:
1.This paper addresses a real and underexplored problem - reasoning shortcut is NeSY, which contributes to the growing field of trustworthy and interpretable AI.
2.There are clear empirical improvements across multiple tasks, and three different NeSy frameworks (DPL, SL, LTN).
3.This paper provides a discussion regarding the limitations, and where the method fails (e.g., in BDD-OIA, a dataset requiring temporal/causal reasoning).

Weakness:
Introduction:
1.The “two-fold intuition” presented as the conceptual motivation lacks clarity and rigor. The first "intuition" is essentially a weak assumption (that a user who writes rules knows at least one example), and the second is a methodological design choice rather than a compelling motivation.
2.The introduction leans heavily on citations to establish context but does not clearly clarify the conceptual novelty of the proposed method. Terms such as semantic alignment and background knowledge satisfaction are introduced early but remain undefined and unquantified.

Method:
1.The integration of prototypes appears empirically motivated but conceptually superficial. While prototypical networks are effective in metric-based few-shot learning, symbolic reasoning often involves non-metric, compositional, and hierarchical structures. The paper does not address how these characteristics are preserved by the use of prototypes.
2.The initialization of prototypes for unspecified classes (Eq. 3) using Gaussian noise added to known centroids is ad hoc and lacks semantic grounding. There is no assurance that these generated prototypes meaningfully represent logically coherent or distinguishable concepts.
3. A closely relevant work is missing: Visual Recognition with Deep Nearest Centroids, ICLR.
4.The dependence on augmentation also undermines generality. Despite claiming that “just one example per class is sufficient,” the method relies heavily on aggressive data augmentation (e.g. 200+ augmented samples per concept). This contradicts the narrative of label efficiency.

There remains a significant conceptual gap between the metric-based mechanism (prototype proximity) and the symbolic reasoning objectives. The assumption that proximity to a prototype leads to logical consistency is not theoretically grounded. The paper would benefit from a clearer, step-by-step formal justification of how and why prototype alignment improves symbolic constraint satisfaction. The paper in its current form feels more like a practical engineering solution effective in constrained synthetic settings, rather than a fundamental advance in neuro-symbolic reasoning.

Evaluation:
1.The performance gains reported (e.g. MNIST-EvenOdd F1 jump from 5% to 98%) are implausibly large. This raises concerns about the simplicity or synthetic nature of the benchmark tasks. Such gains suggest that these tasks may not meaningfully test for generalizable reasoning.
2.The impressive results are confined to synthetic, small-scale benchmarks. The far more modest gains on the more realistic BDD-OIA dataset (F1 from 0.04 to 0.13) suggest that the method's effectiveness severely diminishes when faced with real-world ambiguity, visual complexity, and the need for richer contextual reasoning.
3.The use of the term reasoning is overly broad. The benchmarks primarily involve basic logical structures such as addition or first-order logic. It's unclear how it would help with more complex, multi-step, or abstract reasoning tasks where the concepts themselves are not easily represented by a single visual prototype.
4.The baselines are dated: DeepProbLog (2018), Semantic Loss (2018), and Logic Tensor Networks (2016). The absence of comparisons with more recent NeSy or hybrid neuro-causal models weakens the empirical case for the method’s state-of-the-art relevance.

---

> ### Author Rebuttal · Authors · 2025-07-30
>
> We thank the reviewer for their insightful comments, as we believe implementing them will strongly improve the narrative of the paper.
>
> Due to space concerns, at times we had refer to answers to other Reviewers.
>
> > [Q1+W.Method1] Prototype proximity and logical consistency
>
> Taking into account the proximity of the embeddings of known datapoints when updating the weights of the neural network does not help in better satisfying the constraints, but rather in better assigning the ground truth label to the unlabeled datapoints. Indeed, for many datasets satisfying the constraints is not enough to assign the correct labels.
>
> To give a simple example, suppose we have an image multi-label classification problem with three classes [$Zebra, Mammal, Fish$] and the constraint $Zebra \to Mammal$, stating that if for an image the label $Zebra$ is predicted so should the label $Mammal$.
>
> Suppose we have a dataset with 10 images, 7 portraying fish, 1 portraying a zebra (and hence a mammal) and 2 portraying mammals that are not zebras.
>
> Suppose we have 0 labelled datapoints, then a model always predicting the labels [$Zebra, Mammal$] would achieve perfect logical consistency (and NeSy loss = 0) but it would get accuracy over the labels equal to:
>
> Acc($Fish$) = 0.3, Acc($Zebra$) = 0.1 and Acc($Mammal$) = 0.3
>
> Suppose now we have the same dataset with two labelled datapoints: one portraying a fish and one portraying a zebra (and the rest unlabelled). Then, a model always predicting [$Zebra, Mammal$] for all datapoints, but the one labelled [$Fish$]—for which it predicts [$Fish$]---would achieve loss = 0, but only get accuracy equal to:
>
> Acc($Fish$) = 0.4, Acc($Zebra$) = 0.2 and Acc($Mammal$) = 0.4
>
> Now, if we suppose that the datapoints are clusterable in the embedding space (for a formal analysis see answer to [Q4], Reviewer cxP2), then given the same dataset, a model like ours, if it achieves perfect loss, then it also achieves Acc($Fish$) = Acc($Zebra$) = Acc($Mammal$) = 1.0.
>
> > [Q2] Relationship between semantic alignment and symbolic satisfaction
>
> By semantic alignment we mean associating to each datapoint the right set of labels. For symbolic satisfaction we mean that the set of labels associated to each datapoint satisfies the constraints. Hence, semantic alignment is a sufficient condition for symbolic satisfaction, and symbolic satisfaction is a necessary condition for semantic alignment. We will make this clearer in the final version of the paper.
>
> > [W.Intro1] Intuition lacks rigour
>
> We thank the reviewer for raising this point.
>
> We acknowledge that the first intuition, concerning the availability of randomly annotated datapoints, is indeed an assumption. However, we believe that it is *realistically grounded* in many real-world settings, where practitioners can easily annotate a few datapoints. Thus, our assumption reflects a standard partial supervision regime in which the user is asked to label in most cases just one or at most a handful of datapoints.
>
> Regarding the second intuition, we emphasize that this is not merely a heuristic design choice but a key explanatory mechanism underlying the success of our method, particularly in contrast to prior NeSy approaches.
>
> To illustrate this, consider the MNIST-EvenOdd task discussed in the paper. In this setting, standard NeSy techniques typically proceed as follows:
> 1. For datapoints labelled only with the value of the sum, the model is trained to ensure that the sum of the predicted digits return the correct value
> 2. For datapoints labelled with both individual digit values, the standard cross-entropy loss is applied
>
> A core limitation of this approach lies in the nature of the constraint space. In MNIST-EvenOdd, given the pairs present in the dataset, there are 49 distinct combinations of digit values that satisfy the sum constraint (see answer to [Q4] to reviewer bWcR for a detailed explanation). Hence, for the unlabelled datapoints the model is not guided toward correct digit-level predictions, but is only trained to satisfy the constraint, allowing for many incorrect configurations that are constraint consistent.
>
> The model should instead prefer constraint-satisfying configurations that are similar to those observed in the fully-labelled examples, but, more importantly, this notion of similarity cannot be easily defined in the raw input space, as it would require specifying a highly task-specific metric. Instead, we exploit the learned embedding space—induced by prototypical networks—where similarity to labelled examples emerges naturally through training.
>
> This mechanism allows every datapoint to contribute to training in a way that is guided by proximity to the few annotated examples. As shown in Figure 3 (RQ3), this mitigates the sensitivity to the number of unlabelled datapoints.
>
> We will include the above in the introduction.
>
> >[W.Intro2] Conceptual novelty
>
> The mitigation strategies proposed so far and against which we compare are as follows:
>
> 1. In [1], the authors propose to densely annotate a subset of the labels, and they are able to theoretically prove that if some concepts are fully annotated then the number of reasoning shortcuts a perfectly trained model can take goes down, as expected.
> 2. Again in [1], the authors propose to use a reconstruction loss in addition to the cross-entropy loss. To use this loss, encoder-decoder architectures are needed: the encoder outputs both the concepts and a representation of the input, which is then used by the decoder to reconstruct the input.
> 3. In [2], the authors propose to use the Shannon entropy loss as an additional term in the loss function. Notice that this is equal to 0 only when each distribution $p_\theta(\mathcal{C^k})$ is uniform, which is often not the case.
>
> When compared with the solutions above, ours:
> 1. Does not require dense annotations
> 2. Does not require an encoder-decoder architecture
> 3. Does not assume that the distribution over each $\mathcal{C}^k$ is uniform
>
> Additionally, as shown in Table 2 of the paper, our method greatly outperforms all of the above.
>
> We will make these aspects clearer in the introduction.
>
> > [W.Method1] Not clear how structures are retained in prototypical networks
>
> In our approach, prototypes serve to anchor concept embeddings, while the NeSy loss $\mathcal{L}_{\text{NeSy}}$ encodes the hierarchical and compositional structure and drives the updates of both embeddings and prototypes accordingly.
>
> Section 4 shows that the gradient of the NeSy loss with respect to an embedding $z_I^k$ (for the $k$-th  group $\mathcal{C}_k$) is a weighted combination of prototype directions, where the weights correspond to logic-consistent error signals (Theorem 4.1, Corollary 4.2).
>
> > [W.Method2] Initialization is ad-hoc and lacks semantic grounding
>
> Centroids are only initialized this way and then we train them. They will move according to the knowledge satisfaction: which will give them the semantic grounding.
>
> > [W3.Method3] Missing citation
>
> We thank the reviewer for this citation, we will add it to our discussion. We will also take inspiration from it for our future work, where we will allow for more prototypes for each class.
>
> > [W.Method4] Reliance on data augmentation
>
> Data augmentation is not a necessary aspect of our algorithm, just something that, when possible, leads to more stable centroids. To underline this, in the autonomous driving experiment, we have not used any data augmentation, still getting remarkable improvements. See answer to [Q4] of Reviewer bWcR for more details.
>
> Please notice that data augmentation is a standard technique used in both semi-supervised learning and zero-shot learning, and it upholds the label efficiency narrative, as these annotations are not manually made but rather obtained automatically.
>
> > [W.Evaluation1] Perfomance gains too large, concerns on complexity of the task.
>
> We would like to clarify that the MNIST-EvenOdd and Kand-Logic tasks (together with BDD-OIA) were not introduced by us; rather, they were recently proposed specifically to address the challenge we study. These tasks are part of the “reasoning shortcut benchmark” (rs-bench), introduced at **NeurIPS 2024** (D&B track), with the goal of evaluating the ability of neurosymbolic methods to overcome the reasoning shortcuts problem. Please see answer to [Q4] for reviewer bWcR for more details.
>
> > [W.Evaluation2] More modest results on BDD-OIA
>
> To better highlight how much we can improve even on this task we re-run the experiments on BDD-OIA, but this time we used prototypes for *all concepts*, and not only those that lead to the action Stop. Please see answer to [Q4] Reviewer bWcR for more details on this experiment. The results show that our method can outperform all the considered models.
>
> > [W.Evaluation3] Unclear how the method performs on tasks where concepts are not easily represented by one prototype.
>
> This is surely an interesting question for future work. However, regarding the current paper, please take into account that we manage to significantly improve the results on the the most recent benchmark proposed to evaluate NeSy models, where current methods do not manage to disambiguate between concepts that can be represented as a single prototype.
>
> > [W.Evaluation4] Old baselines
>
> The baselines still represent sota methods in NeSy, they are used in many papers studying the reasoning shortcuts problems (e.g., rs-bench), and we compared against mitigation strategies that are more recent [1,2].
>
> In answer to [Q1+W1.b] to Reviewer CSxp we showed that our method is able to greatly outperform ABLKit as well.
>
> ### References
>
> [1] Marconato et al Not All Neuro-Symbolic Concepts Are Created Equal: Analysis and Mitigation of Reasoning Shortcuts, NeuIPS, 2023
>
> [2] Manhaeve et al Neural probabilistic logic programming in deepproblog. Artificial Intelligence, 2021
>
> [3] Yu-Xuan Huang et al., ABLkit: a Python toolkit for abductive learning, Frontiers of Computer Science, 2024

---

> > ### Comment · Reviewer_wXk3 · 2025-08-05
> > **Thanks for the response**
> >
> > I thank the authors for their response. While some concerns have been addressed, this work still requires significant improvement to meet NeurIPS standards. It is not yet ready for publication, as substantial additional work is needed. I will maintain my original score.

---

> > > ### Author Response · Authors · 2025-08-05
> > >
> > > Dear Reviewer,
> > >
> > > Thank you for engaging with us.
> > >
> > > Could you please let us know which concerns we have not yet addressed?
> > >
> > > A more precise list will help us improve our work also for the future.

---

> ### Comment · Area_Chair_w3ks · 2025-08-05
> **Ping**
>
> Dear Reviewer,
>
> The deadline for the author-reviewer discussion is approaching (Aug 8, 11.59pm AoE).
> Please read carefully the authors' rebuttal and engage in meaningful discussion.
>
> Thank you,
> Your AC

---

### Official Review · Reviewer_bWcR · 2025-07-05

**Clarity:** 1
**Significance:** 2
**Originality:** 3
**Rating:** 3
**Confidence:** 4

**Summary:**

This paper introduces a method called PNet for utilizing limited labeled data in Neuro-Symbolic tasks under the guiding principle of constructing class prototypes. Specifically, for classes with labeled data, the mean of the (augmented) embeddings is used as the class prototype; for classes without labeled data, a prototype is randomly initialized based on the distances between existing prototypes. By doing so, this paper introduces an additional loss term based on the distance between embeddings and prototypes, alongside the traditional NeSy loss. This approach balances logical constraints and label information, mitigating the issue of reasoning shortcuts. Authors conducted multiple experiments to validate the effectiveness of the proposed method.

**Questions:**

1. Why does the prototype selection for classes without labeled data require introducing randomness (Equation (3))?
2. Since this paper considers a classic neuro-symbolic task, it would be beneficial to adopt a more concise and traditional set of symbols to enhance the readability.
3. If the proposed method can demonstrate a significant performance advantage in the fair comparisons mentioned in the weaknesses, I would consider increasing the score.
4. If the method can showcase its superiority in more challenging scenarios, I would consider increasing the score.

**Ethical Concerns:**

["NO or VERY MINOR ethics concerns only"]

**Final Justification:**

The authors address some of my concerns. Hence I increase my score. I suggest the authors further improve experiments and writing.

**Limitations:**

yes

**Quality:**

2

**Strengths And Weaknesses:**

Strength:

1.Authors considered multiple factors that could affect the effectiveness of the method and demonstrated its performance under different conditions through experiments.

2.Authors demonstrated the limitations of the method through experiments.

Weakness:

1.The main experiment considered in this paper is overly simplistic—a (minor) variant of MNIST add with a small amount of labeled data. It is well-known that simple CNN networks can achieve over 95% accuracy on the MNIST dataset even when each class contains only one labeled image. Therefore, to validate the effectiveness of the proposed method, a baseline that first performs supervised learning using labeled data and then conducts neuro-symbolic learning should be added.

2.The comparison between this method and others is not entirely fair. Other methods target more challenging scenarios without labeled data, whereas in cases where some labeled data is available for supervised pretraining, I believe the comparison methods could achieve similar performance to the proposed method.

3.The paper addresses a classic neuro-symbolic task but introduces a new and relatively complex set of symbols in Section 2, particularly merging the Concept space and Final label space, which makes it difficult to read.

4.The paper contains several typos and imprecisions, such as: two instances of "(iii)" in Section 2; the definition of mm on line 87 is not a mapping to the power set, which contradicts Example 2; and Equation (3) includes two $C_c$ symbols with entirely different meanings, further complicating readability.

---

> ### Author Rebuttal · Authors · 2025-07-30
>
> We would like to thank the reviewer for their thoughtful and constructive comments. We will incorporate the suggested discussions and clarifications in the revised manuscript, which will strengthen the overall contribution and improve the presentation and interpretation of the experimental results.
>
> > [Q1] Why randomness?
>
> We chose to retain the well-known benefits of random weight initialization in deep learning—such as effective exploration of the parameter space and robustness across different seeds. In our setting, centroids act as learnable parameters (analogous to weights) and thus benefit from similar initialization strategies.
>
> However, initialization is particularly important in our setting due to the presence of reasoning shortcuts. Without any control on their placement, a prototype may be placed far from the relevant embeddings, leading the model to ignore it and instead collapse multiple concepts onto a single, incorrect prototype that still satisfies the constraints. To mitigate this, we sample centroids within a hyperball centered on the mean of known prototypes (with $p=0.99$). This encourages separation among mutually exclusive classes and reduces shortcut-induced failures, while still preserving the flexibility and diversity of random initialization.
>
> > [Q2] Standard notation.
>
> If we followed the standard notation used e.g., in [1] then we would have made the assumption that there are a set of concepts (or intermediate outputs that are not supervised) and a set of final labels (of final outputs which are fully supervised). However, this assumption is not necessary in many NeSy systems (like SL), as they can seamlessly handle scenarios where each datapoint can be partially labelled with any subset of the labels, and there is not inherent distinction between intermediate concepts and final labels.
>
> Additionally, the current notation allows us to explicitly associate every atom in the formula with a specific input-output pair of the neural network, making the connection between the two more explicit.
>
> > [Q3] Unfair comparison
>
> Regarding this point we would like to highlight that, in addition to the standard baselines (with no access to labelled datapoints)---indicated with LTN, DPL and SL, we have also compared with:
> 1. *Fully supervised CNNs (the same we used as prototypical extractors) that have access to the same amount of supervised datapoints* on the MNIST-EvenOdd task, as reported with “Standard NN” in Figure 3. We agree that we should have run this baseline also on Kand-Logic and we should have included these results in Table 1. We will do so in case of acceptance. Below, we report the performance on the MNIST-EvenOdd dataset in Table A and the performance on Kand-Logic in Table B. As we can see from the baseline, with the small augmented labelled dataset available to our PNets, the fully supervised baselines only managed to get mean accuracy equal to 0.59 on MNIST-EvenOdd and 0.34 on Kand-Logic. We have also run the fully supervised baseline on the autonomous driving dataset (BDD-OIA), see answer below.
> 3. *Augmented neurosymbolic AI baselines which are trained with both labelled and unlabelled datapoints* indicated with LTN(Aug), DPL(Aug) and SL(Aug) - these methods are  trained with exactly the same dataset as our methods. As we can see from Table 1, adding the same amount of supervisions helps in improving their performance, but in a much more limited way than using the PNets. This is because, for each unlabeled datapoint, the weights are updated solely based on the degree of constraint satisfaction. In contrast, when using prototypical networks, the update also accounts for the proximity of the embedding to each prototype. This also explains why our method exhibits significantly greater robustness to the number of unlabeled datapoints used during training compared to standard architectures, as demonstrated in RQ3 of the paper.
>
> Table A: Performance on MNIST-EvenOdd for (i) the fully supervised baseline, (ii) the augmented NeSy methods and (iii) our NeSy + prototypical networks. For F1 the higher the score the better---indicated as ($\uparrow$)---for Cls the lower the score the better---indicated as ($\downarrow$). Best results are always in bold
>
> | Task|Acc(C) ($\uparrow$) | F1(C) ($\uparrow$)|Cls(C) ($\downarrow$)|
> |--|--|--|--|
> | Fully Supervised| 0.59±0.02 | 0.55±0.02 | 0.01±0.01|
> | LTN(Aug)   | 0.46±0.31| 0.37±0.35| 0.49±0.29|
> | LTN+PNet(Ours)| **0.98±0.00**| 0.97±0.00| **0.00±0.00** |
> | SL(Aug)|0.56±0.24| 0.45±0.27|0.26±0.24|
> | SL+PNet(Ours)|**0.98±0.00**|**0.98±0.01**| **0.00±0.00**|
> | DPL(Aug)|0.26±0.11|0.15±0.07| 0.68±0.06|
> | DPL+PNet(Ours)|**0.98±0.00**|**0.98±0.00**|**0.00±0.00**|
>
> Table B: Performance on Kand-Logic for (i) the fully supervised baseline, (ii) the augmented NeSy methods and (iii) our NeSy + prototypical networks
>
> |Task|Acc(C) ($\uparrow$)|F1(C) ($\uparrow$)|Cls(C) ($\downarrow$)|
> |--|--|--|--|
> |Fully Supervised|0.34±0.01|0.34±0.01| **0.00±0.00** |
> |LTN(Aug)|0.33±0.01|0.31±0.02|0.59±0.17
> |LTN+PNet(Ours)|**0.94±0.00**|**0.94±0.00**|**0.00±0.01** |
> |SL(Aug)|0.36±0.02| 0.22±0.05|0.78±0.11|
> |SL+PNet(Ours)|**0.94±0.01**|**0.93±0.01** |**0.00±0.02**|
> |DPL(Aug)| 0.35±0.01|0.26±0.00|0.70±0.07|
> |DPL+PNet(Ours)|**0.94±0.00**|**0.94±0.00**|**0.00±0.00**|
>
> We believe this comparison is fair, as every model has access to the same amount of data.
>
> > [Q4] Simple experiments
>
> We would like to clarify that the MNIST-EvenOdd and Kand-Logic tasks were not introduced by us; rather, they were recently proposed specifically to address the challenge we study. These tasks are part of the “reasoning shortcut benchmark” (rs-bench), introduced at **NeurIPS 2024** (D&B track), with the goal of evaluating the ability of neuro-symbolic methods to overcome the reasoning shortcuts problem.
>
> These tasks are particularly challenging due to the large number of possible reasoning shortcuts a model can exploit. For example, in the MNIST-EvenOdd task, there are 49 different digit-to-label assignments that perfectly satisfy the logical constraints, yet only one of them corresponds to the correct digit labeling. This arises because only a limited set of digit pairs appears in the training set. In Table C below we give the list of the 8 possible input image pairs together with their associated label (their sum) appearing in the dataset:
>
> Table C. Input image pairs and their labels in MNIST-EvenOdd dataset
> |Possible Input Image Pairs| Label|
> |--|--|
> |(0,6)|6
> |(2,8)|10
> |(4,6)|10
> |(4,8)|12
> |(1,5)|6
> |(3,7)|10
> |(1,9)|10
> |(3,9)|12
>
> As a result, the model may converge to a spurious mapping that satisfies the sum constraint on all training examples but mislabels every digit. For instance, interpreting each 0 image as 4 and 6 image as 2 leads to correctly infer the label 6 for every image pair (0,6). Similarly, the interpretation in Table D achieves perfect constraint satisfaction on the training set while failing to correctly classify every single digit image:
>
> Table D. Possible digits assignment for MNIST-EvenOdd
>
> |Actual Digit in the Image| Digit assigned by the model |
> |--|--|
> |0 | 4
> |1 | 5
> |2 | 6
> |3 | 7
> |4 | 8
> |5 | 1
> |6 | 2
> |7 | 3
> |8 | 4
> |9 | 5
>
> Hence, even though MNIST-EvenOdd and Kand-Logic might appear simple datasets, they are actually not for NeSy models, as the knowledge is not enough to correctly identify the concepts. Additionally, as we have seen in the previous question, few labelled datapoints are *not* enough to achieve ~0.90 accuracy.
>
> However, we agree that real world scenarios might present noisier data. To better highlight how our model would perform in noisy and real world scenarios, we re-run the experiments on the autonomous driving dataset (BDD-OIA), but this time we used prototypes for *all concepts*, and not only those that lead to the action Stop.
>
> To have a fair comparison, we also
> 1. trained DPL not only with the unlabelled data but also with the 128 datapoints available to our method (indicated as DPL (Aug)), and
> 2. the model in a fully-supervised way (using all the 16k datapoints), thus highlighting the difficulty of the task.
>
> We report the overall results in Table E.
>
> Table E. Results on the autonomous driving dataset (BDD-OIA)
>
> |Model|F1(C) ($\uparrow$)|Cls(C) ($\downarrow$)|
> |--|--|--|
> |DPL|0.04±0.00|0.85±0.01|
> |DPL (Aug)|0.07±0.07|0.75±0.05|
> |Fully Supervised|0.06±0.06|0.78±0.10|
> |DPL+PNet|**0.16±0.15**|**0.35±0.01**|
>
> Finally, we report the results when stratifying the concepts according to the resulting action in Table F. For example, F1(Forward) is calculated as the mean over the F1 of all the concepts that cause the action ''Forward'', i.e., "Green Light", "Follow" and "Road Clear".
>
> Table F: Results on BDD-OIA stratified by the observed action
> |Method| F1(Forward)|Cls(Forward)|F1(Stop)|Cls(Stop)|F1(Left)| Cls(Left)| F1(Right)| Cls(Right)|
> |--|--|--|--|--|--|--|--|--|
> |DPL (Aug)|0.19±0.12| 0.80±0.09 |0.08 ±0.08|0.87±0.03|0.01±0.01|0.50±0.00|0.01±0.01|0.84±0.08|
> |Fully Supervised|0.15±0.15|0.78±0.12|0.04±0.03|0.89±0.05|0.02±0.01|0.50±0.24|0.02±0.02|0.95 ±0.02|
> |DPL+PNet|**0.35±0.07**|**0.23±0.23**|**0.09±0.07**|**0.40±0.11**| **0.08±0.02**|**0.00±0.00**|**0.17±0.01**|**0.77±0.09**|
>
> Our original aim in including the BDD-OIA experiment (over just some concepts) was to demonstrate that our approach remains effective even in more realistic and noisy scenarios, where perfect accuracy—as seen in MNIST-EvenOdd and Kand-Logic—is not attainable. However, we acknowledge that the current presentation may have overemphasized the limitations without sufficiently highlighting the strengths. To address this, we will include the above additional results covering all concepts as well as comparisons with the fully-supervised baseline, which will better showcase the model's robustness and generalization in this setting.
>
> ### References
>
> [1] Marconato et al. Not All Neuro-Symbolic Concepts Are Created Equal: Analysis and Mitigation of Reasoning Shortcuts, *NeurIPS*, 2023

---

> > ### Comment · Reviewer_bWcR · 2025-08-05
> > **Thank you**
> >
> > Dear authors,
> >
> > Thank you for your response. First, I would like to clarify a point. The simplicity of the task that I referred to focuses on the fact that there is some labeled data, not that the task is simple in the absence of labeled data, as mentioned in the weaknesses. Secondly, the experiment I mentioned, which involves pre-training with supervised data first and then using the baseline algorithm, does not seem to appear in the rebuttal. I prefer keeping my score.

---

> ### Comment · Area_Chair_w3ks · 2025-08-05
> **Ping**
>
> Dear Reviewer,
>
> The deadline for the author-reviewer discussion is approaching (Aug 8, 11.59pm AoE).
> Please read carefully the authors' rebuttal and engage in meaningful discussion.
>
> Thank you,
> Your AC

---

> ### Author Response · Authors · 2025-08-05
>
> Dear Reviewer,
>
> We thank for engaging with us.
>
> We would like to point out that all the LTN(Aug), SL(Aug) and DPL(Aug) are trained with all the available supervised data and  unsupervised datapoints. The difference in performance between our method and these methods shows that the presence of the labelled datapoints does not make the task easy.
>
> We understood now that the reviewer meant that we should have pre-trained until convergence on the supervised data and then fine-tune on the unsupervised. We are happy to provide this experiment. Just allow us some time to run it.
>
> Thanks a lot for your time.

---

> > ### Author Response · Authors · 2025-08-06
> >
> > Dear Reviewer,
> >
> > As promised, we have run the experiments with pre-training on supervised data and then using the baseline algorithm.
> >
> > We indicate the models trained this way with *(Nesy-baseline)-Pre* in the tables below.
> >
> > In Table A we report the results obtained on MNIST-EvenOdd and in Table B we report the results obtained on BDD-OIA.
> >
> > Table A. *Results for MNIST-EvenOdd*
> >
> > | Model | F1(C) ($\uparrow$) | Cls(C) ($\downarrow$) |
> > |--|--|--|
> > |DPL-Pre| 0.42±0.26 | 0.46±0.23 |
> > | DPL+PNet|  **0.98±0.00** | **0.00±0.00** |
> > |LTN-Pre| 0.36±0.12 | 0.51±0.12 |
> > |LTN+PNet| **0.97±0.00**  | **0.00±0.00** |
> > |SL-Pre| 0.37±0.19 | 0.51±0.17 |
> > |SL+PNet| **0.98±0.01** | **0.00±0.00** |
> >
> >  Table B. *Results on BDD-OIA*
> >
> > | Model | F1(C) ($\uparrow$) | Cls(C) ($\downarrow$) |
> > |--|--|--|
> > |DPL-Pre| 0.06±0.06 | 0.79±0.01
> > |DPL+PNet | **0.16±0.15** | **0.35±0.01** |
> >
> > As we can see from the results, our model **clearly outperforms** the pre-trained ones.
> >
> > We hope this answer solved all concerns, and you might consider raising the score.
> >
> > Please let us know if you have any outstanding concerns, we are happy to try and resolve them.

---

> > > ### Author Response · Authors · 2025-08-08
> > >
> > > Dear Reviewer bWcR,
> > >
> > > as the discussion period is coming to an end, we would be extremely grateful if you could consider our latest answer to your comments, as it shows that our method **clearly outperforms models pre-trained with labelled data both in simple and challenging scenarios**.
> > >
> > > We thank you in advance for your time and we remain available for any other question.

---

> > > > ### Comment · Reviewer_bWcR · 2025-08-09
> > > > **Thank you**
> > > >
> > > > Dear Authors,
> > > >
> > > > Thank you for the explanations and additional experiments. However, the experimental results appear questionable to me. As you mentioned in the main text, in the MNIST-EvenOdd experiment, you randomly select and label one image for each class. This is a very strong assumption for MNIST-related experiments; these labels are sufficient for a CNN to learn effective features, and with the help of the Neuro-Symbolic approach, the model is able to achieve a classification accuracy of over 95%.
> > > >
> > > > Additionally, the paper's symbol set still needs to be simplified to improve readability. I will update my score, but the paper still requires significant improvements to meet the NeurIPS standard.

---

### Note · Authors · 2025-08-12

Dear AC and dear Reviewers,
thank you for engaging with us and for the useful comments.
However, we would like to clarify some points:
1. While we do not use the traditional set of symbols, we consider this choice a way of conveying
an original perspective in stating the problem: indeed, by dropping the standard notation, we
    $\textbf{(a)}$ bridge the gap between intermediate concepts and labels and
    $\textbf{(b)}$ associate every atom in the constraints with an input-output pair,
        highlighting their connection more clearly.
$\textit{This is crucial to understand the nature of the reasoning shortcut problem we are addressing}$.

2. Some concerns were raised about the synthetic nature of tasks such as
MNIST-EvenOdd and Kand-Logic. However [1] shows that the baselines we consider greatly
struggle with learning the concepts for these tasks already. Moreover, to solve all the Reviewer's concerns, we significantly expanded our experimental analysis for the challenging, high-stake task, BDD-OIA and showed the proposed approach clearly outperforms the (pretrained and supervised) baselines.

$\textit{This makes evaluation exhaustive}.$

3. We believe that the additional results we provided to solve
the concerns from Reviewer bWcR were not completely understood. While the annotations we
use are sufficient for a CNN in general,
it is not the case for the MNIST-EvenOdd variation. To see this, consider the results for
the mitigation strategy in [2] that provides the baselines with supervisions for
$\textbf{all}$ digits 4 and 9 (Table 3): crucially, the F1 score for digits is still $\textbf{0.1}$
and $\textbf{0.2}$ for DPL and LTN. Hence, the results we provided, obtained
using annotations to
    (i) provide $\textbf{supervisions}$ (as [2] describes) and
    (ii) $\textbf{pretraining}$ (as the reviewer pointed out)
are confirming the results already in the literature.

$\textit{For this reason please observe that our comparison is not questionable or unfair}$.

4. We acknowledged that Reviewer wXk3 required additional work, but they did not explain
which concerns were not addressed during the rebuttal. We tried to reach out for clarification,
but did not receive a response.

Thank you,

The Authors.


$\textit{References}$:

[1] $\textit{A Neuro-Symbolic Benchmark Suite for Concept Quality and Reasoning Shortcuts}$,

[2] $\textit{Not All Neuro-Symbolic Concepts Are Created Equal: Analysis and Mitigation of Reasoning Shortcuts}$

---

### Decision · Program_Chairs · 2025-09-17

**Decision:**

Accept (poster)

**Comment:**

All reviewers agree that the paper tackles a significant problem in neuro-symbolic integration. Most of the issues raised during in the reviewers were addressed by the rebuttal, and in fact two reviewers increased their score. The other two were not particularly responsive. I went through their main gripes with the paper (such as usage of simple data sets, surprisingly large performance gains, lack of fairness in the evaluation), and it seems to me they *have* been addressed by the authors.

For this reason, I am leaning toward acceptance. I recommend the authors to clarify all points of contention in the paper and to implement all changes they promised in their rebuttal.